

# Driver versus navigator causation in biology: the case of insulin and fasting glucose

Manawa Diwekar-Joshi[1] and Milind Watve[2]

[1] Biology, Indian Institute of Science Education and Research, Pune, Maharashtra, India
[2] Deenanath Mangeshkar Hospital and Research Centre, Pune, Maharashtra, India

## ABSTRACT

**Background:** In biomedicine, inferring causal relation from experimental intervention or perturbation is believed to be a more reliable approach than inferring causation from cross-sectional correlation. However, we point out here that even in interventional inference there are logical traps. In homeostatic systems, causality in a steady state can be qualitatively different from that in a perturbed state. On a broader scale there is a need to differentiate driver causality from navigator causality. A driver is essential for reaching a destination but may not have any role in deciding the destination. A navigator on the other hand has a role in deciding the destination and the path but may not be able to drive the system to the destination. The failure to differentiate between types of causalities is likely to have resulted into many misinterpretations in physiology and biomedicine.
**Methods:** We illustrate this by critically re-examining a specific case of the causal role of insulin in glucose homeostasis using five different approaches (1) Systematic review of tissue specific insulin receptor knock-outs, (2) Systematic review of insulin suppression and insulin enhancement experiments, (3) Differentiating steady state and post-meal state glucose levels in streptozotocin treated rats in primary experiments, (4) Mathematical and theoretical considerations and (5) Glucose-insulin relationship in human epidemiological data.
**Results:** All the approaches converge on the inference that although insulin action hastens the return to a steady state after a glucose load, there is no evidence that insulin action determines the steady state level of glucose. Insulin, unlike the popular belief in medicine, appears to be a driver but not a navigator for steady state glucose level. It is quite likely therefore that the current line of clinical action in the field of type 2 diabetes has limited success largely because it is based on a misinterpretation of glucose-insulin relationship. The insulin-glucose example suggests that we may have to carefully re-examine causal inferences from perturbation experiments and set up revised norms for experimental design for causal inference.

Corresponding authors
Manawa Diwekar-Joshi,
manawa.diwekar@students.
iiserpune.ac.in
Milind Watve,
milindwatve@gmail.com

## INTRODUCTION

### Causal relationships in biology

Inferring causal relationships in biology is a more complex, philosophical and methodological issue than what is generally perceived. Much of the debate has centred around making causal inferences from observed associations or correlations (*Boudon, 1965*; *Chawla et al., 2018*; *Grace, 2002*; *Hill, 1965*). Methods of analysis specifically addressing the question of causal inferences have been developed for various contexts and at various levels of reasoning including Hill criteria (*Hill, 1965*), path analysis (*Li, 1956*; *Niles, 1923*; *Wright, 1960*), Granger causality (*Granger, 1969*), steady state causality (*Chawla et al., 2018*), genomic and network causality (*Kulkarni, Sharda & Watve, 2017*; *Meinshausen et al., 2016*; *Triantafillou et al., 2017*). While on the one hand, complex statistical and computational tools are being developed for determining causality under different contexts, certain simple problems about causality are not sufficiently appreciated and addressed. We focus on one such apparently simple problem here, the lack of appreciation of which has led to serious misinterpretations of experimental results.

In classical experimental physiology, interventions or perturbations are believed to be reliable indicators of causation and there is little debate about it. If the experimenter perturbs A and finds a significant effect on B after following all fundamental principles of experimental design, the change in A is inferred to be causal to the change in B. However, there are many subtleties in drawing a causal inference from experimental interventions that have not yet attracted sufficient philosophical as well as methodological attention among experimental biologists. One such thinking trap is that in homeostatic systems the nature of causality in a perturbed state can be qualitatively different from that in equilibrium or steady state and the failure to distinguish between the two may have substantially misled biomedical research.

### The role of growth rates in Lotka-Volterra competition models

A well worked out theoretical model that can be used to distinguish clearly between perturbed and steady state causation is the Lotka-Volterra (LV) competition model. This model describes the dynamics for interspecific competition (*Gotelli, 2008*). The growth of two interacting populations is modelled using logistic equations for both the populations. The changes in populations depend on the individual growth rates and the carrying capacities of the two populations (Eqs. 1 and 2). Additional parameters are the competition coefficients α and β which represent the effects of the two species on each other. Thus the carrying capacities $K_1$, $K_2$ and the competition coefficients α and β determine the equilibrium population (Eqs. 3 and 4) (*Gotelli, 2008*).

$$\frac{dN_1}{dt} = r_1 . N_1 \left( \frac{K_1 - N_1 - \alpha . N_2}{K_1} \right) \tag{1}$$

$$\frac{dN_2}{dt} = r_2 . N_2 \left( \frac{K_2 - N_2 - \beta . N_1}{K_2} \right) \tag{2}$$

$$\check{N}_1 = \frac{K_1 - \alpha.K_2}{1 - \alpha.\beta} \qquad (3)$$

$$\check{N}_2 = \frac{K_2 - \beta.K_1}{1 - \alpha.\beta} \qquad (4)$$

where,

$N_1$ and $N_2$ are the population sizes of the two competing species respectively, $\check{N}_1$ and $\check{N}_2$ representing steady state populations.

$r_1$ and $r_2$ are the growth rates

$K_1$ and $K_2$ are the carrying capacities

$\alpha$ is the competition coefficient which shows the effect of population 2 on population 1

$\beta$ is the competition coefficient which shows the effect of population 1 on population 2

If, in an experiment, we start at a non-equilibrium state, and observe at time T1 (Fig. 1), the standing populations would be inferred as a function of intrinsic growth rates $r_1$ and $r_2$. But if observed at time T2, the inference would be different. The intrinsic growth rates of both the competing populations do not determine the equilibrium populations of the two species. The magnitude of $r$ determines the time taken by the population to reach the equilibrium or steady state (Fig. 1). The role of growth rates in population dynamics is well recognised and is demonstrable in a perturbed state but it needs to be realised that it has no role in determining the steady state populations. Nevertheless, existence of non-zero positive growth rates is essential for attaining the equilibrium or returning to it if perturbed. If either or both the growth rates are made zero, the system will never attain back a stable equilibrium coexistence. Thus, the two growth rates are causal for attaining equilibrium, but they have no causal role in deciding the position of the equilibrium point. Thus, we need to distinguish between the driver cause and the navigator cause. Driver causality is a process that takes a homoeostatic system to an equilibrium point but may not have any role in deciding the attributes of the equilibrium. Navigator causality refers to the processes that determine the location of the equilibrium point and lead the driver there, but in the absence of the driver, may not be able to take the system to the steady state.

For a homeostatic system, the distinction between perturbed state and steady state causality is practically equivalent to driver and navigator causality. However, the driver-navigator distinction can be applied, in principle, to non-homeostatic systems as well and therefore is a broader concept.

This has relevance to experimental physiology. If knocking out a certain gene, protein or function disables homeostatic control, it does not provide us any clue as to whether it has a driver or navigator function. The experiment does not necessarily demonstrate that the gene, protein or function determines the steady state levels of the controlled variable. Since distinction between driver and navigator causality has not been explicitly made in experimental physiology, currently there are no norms or methods to resolve between the

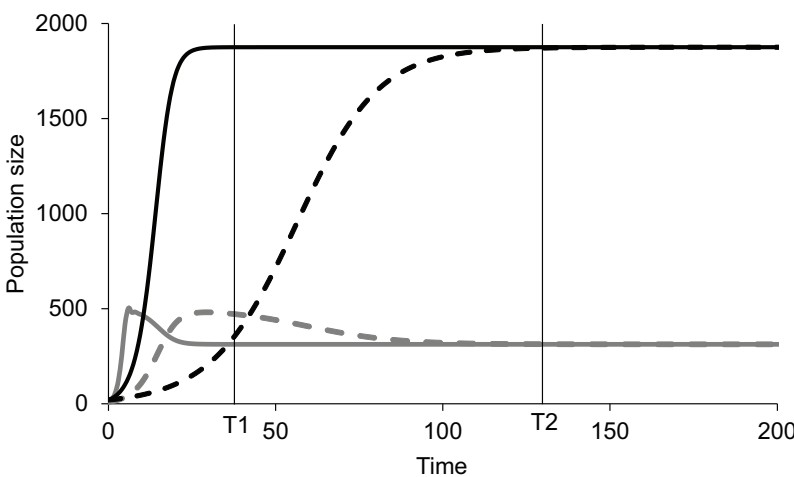

**Figure 1 Lotka-Volterra model of two competing species at different growth rates.** Simulated population dynamics of two competing species A (black lines) and B (grey lines) at different growth rates. Solid lines represent high growth rates reaching the equilibrium faster whereas dotted lines represent low growth rates reaching the equilibrium more slowly. In both cases species A has slower growth rate but greater carrying capacity than species B. At time T1 the populations are in proportion to their growth rates but at T2, approaching equilibrium, growth rates become increasingly irrelevant in determining population sizes. Thus, growth rates are important determinants of population size in a perturbed state but not at a steady state.

two types of causations. We use this distinction below to re-examine the role of insulin in glucose homeostasis and show that the failure to distinguish between driver and navigator causality has led to a fundamentally flawed understanding of glucose homeostasis and type 2 diabetes in particular.

## Why is insulin believed to regulate fasting blood sugar: a burden of history?

After the classical demonstration by Claud Bernard that damage to medulla oblongata causes hyperglycaemia(*Bernard, 1879*), the second major breakthrough was the demonstration by *Von Mering & Minkowski (1890)* that pancreatectomy resulted in hyperglycaemia and further that pancreatic extracts resulted in lowering of plasma glucose. The active principle eventually purified became known as insulin (*Karamitsos, 2011*). The discovery and success of insulin in treating diabetes was so overwhelming that insulin became the key molecule in glucose homeostasis and the role of brain and other mechanisms were practically forgotten. It should be noted that the prevalent type of diabetes then was what we would call type 1 diabetes (T1D) today in which there is almost complete destruction of pancreatic β-cells. The distinction between type 1 and 2 developed gradually over the next five decades along with the realisation that insulin levels may be normal or raised in type 2 diabetes (T2D) and that a substantial population of β-cells survives lifelong (*Butler et al., 2003*; *Clark et al., 1990*; *Porte & Kahn, 2001*). However, by now the thinking about glucose homeostasis was so insulin-centered, that the inability of normal or raised levels of insulin to keep plasma glucose normal was labelled as "insulin resistance"(*Reaven, 1988*) without adequately examining and eliminating alternative

possibilities and the concept got wide uncritical acceptance. Although insulin receptor and downstream functions are known to be highly variable at the cellular level, the question whether altered insulin signalling is solely or mainly responsible for fasting hyperglycaemia of T2D, or other insulin independent mechanisms play a significant role is not clearly answered.

There are multiple reasons to doubt and re-examine the role of insulin in glucose regulation in relation to T2D (*Corkey, 2012*; *Pories & Dohm, 2012*; *Watve, 2013*). Exogenous insulin and other insulin-centered lines of treatment have largely failed to reduce diabetic complications and mortality in T2D although short term glucose lowering may be achieved (*ACCORD, 2008*; *King, Peacock & Donnelly, 2001*; *Meinert et al., 1970*; *UK Prospective Diabetes Study Group, 1998a*, *1998b*, *1998c*). In the long run even the glucose normalisation goal is not achieved in majority of cases (*UK Prospective Diabetes Study Group, 1998b*, *1998c*). A number of mechanisms are known to influence glucose dynamics, partially or completely independent of insulin signalling, including autonomic signals (*Nonogaki, 2000*; *Schwartz, 2005*), glucocorticoids (*Di Dalmazi et al., 2012*; *Gathercole & Stewart, 2010*; *Goldstein et al., 1993*; *Kuo et al., 2015*), insulin independent glucose transporters (*Carruthers et al., 2009*) and certain other hormones and growth factors (*Clemmons, 2004*; *Jansen et al., 2006*; *Messmer-Blust et al., 2012*; *Suh et al., 2014*). Analysis of multi-organ signalling network models have also raised doubts about the central role of insulin and insulin resistance in T2D (*Kulkarni, Sharda & Watve, 2017*).

The definitions as well as clinical measures of insulin resistance are such that the effects of all other mechanisms are accounted for under the name of "insulin resistance". For example, the HOMA-IR index is calculated as a product of fasting glucose and fasting insulin (*Matthews et al., 1985*; *Turner et al., 1979*). The belief that this product reflects insulin resistance is necessarily based on the assumption that insulin signalling alone quantitatively determines glucose level in a fasting steady state. The assumption has seldom been critically examined. If any other mechanisms are contributing to impaired fasting glucose, they will be included in the HOMA-IR index going by the way it is calculated and would be labelled as insulin resistance. This amounts to a circular logic. Insulin resistance is hypothesised to be responsible for the failure of insulin to control fasting glucose, and insulin resistance is measured as the inability of insulin to control fasting glucose. This makes the insulin resistance concept unfalsifiable from clinical data.

We have previously showed using mathematical and statistical tools of causal analysis (*Chawla et al., 2018*) that the classical pathway of obesity induced insulin resistance leading to a hyperinsulinemic normoglycemic prediabetic state and the faithfulness of HOMA indices in measuring insulin resistance cannot be simultaneously true. Either the HOMA indices do not represent insulin resistance faithfully or the classically believed pathway of compensatory insulin response leading to hyperinsulinemic normoglycemic state is wrong according to this analysis (*Chawla et al., 2018*). All the mounting inconsistencies and paradoxes in the insulin resistance concept along with the limited success of type 2 diabetes treatment warrant a re-examination of the foundational concepts.

## Approach used in this paper

We examine here the long-held belief that altered insulin signalling is responsible for fasting as well as post prandial hyperglycaemia in T2D using five different approaches:

(1) Systematic review of experiments involving tissue specific insulin receptor knock-outs (IRKOs);

(2) Systematic review of experiments to chronically raise or lower insulin levels;

(3) Primary experiments on streptozotocin (STZ) induced hyperglycaemia in rats, that differentiate between steady (fasting) and perturbed (post-feeding) state;

(4) Examining the insulin resistance hypothesis for being mathematically possible and theoretically sound;

(5) Analysis of insulin-glucose relationship in steady state versus post-meal perturbed state in human epidemiological data for testing the predictions of mathematical models.

The first three approaches have the advantage of using specific molecular interventions where the target is precisely known. For the meta-analyses we chose mechanisms of insulin level/action modification which have been used extensively and have been reproduced by multiple labs world over. The possible disadvantage is that they are mostly animal experiments and doubts are expressed about whether the results are directly relevant to humans (*Akhtar, 2015*; *Ali, Chandrasekera & Pippin, 2018*; *Bracken, 2009*). However, some of the experiments reported are human and they converge with the inferences of the animal experiments. In the last two approaches, human epidemiological data are used in which the experimental molecular precision is not expected, but we test certain specific predictions of the insulin resistance hypotheses using novel analytical approaches and examine whether they converge on similar inferences. The convergence of human and animal data is important to reach robust conclusions.We will now describe each approach in detail here.

### *Systematic review of experiments involving tissue specific insulin receptor knock-outs*

The first step in insulin signalling is the binding of insulin to insulin receptor (*Bevan, 2001*). The downstream actions of this event finally lead to insulin-dependent glucose uptake in insulin dependent tissues of the body. Experimentally, disruption of insulin signalling is achieved by knocking out or inhibiting various players in the signalling cascade. We chose to look at the effects of knocking out the tissue specific insulin receptor on fasting and post-meal/feeding or post glucose load levels in rodent models. Studying tissue specific insulin receptor knockouts enables us to differentiate between the roles of insulin signalling in different tissues. A classical belief is that the post-meal glucose curve is mainly influenced by the rate of glucose uptake by tissues, mainly muscle, whereas the fasting glucose levels are mainly determined by the rate of liver glucose production (*Bock et al., 2007*). If this belief is true one expects that muscle specific knockout would mainly affect the GTT curve but may not affect fasting glucose level, whereas liver specific knockout would mainly affect the fasting glucose level. We searched the published

literature for experiments in which the fasting and post-feeding levels of glucose are measured in tissue specific insulin receptor knock-outs.

### Systematic review of experiments to chronically raise or lower insulin levels

The insulin receptor knockout experiments assume that the main action of insulin is through the specific receptors. It can be argued that insulin acts through other receptors or may have other mechanisms of action yet unknown and therefore receptor knockouts do not fully eliminate insulin action. Alternatively, we can alter insulin level itself to see how it affects glucose level in fasting state or post glucose load. Insulin is known to alter plasma glucose immediately on administration (*Galloway et al., 1981*), but this is not a steady state response. If insulin levels can be raised or lowered and sustained long enough to reach a steady state, the effect of insulin on glucose in a steady state can be studied. If insulin affects steady state glucose, a sustained rise in insulin will result into a sustained lower steady state glucose level. Conversely a sustained suppression of insulin would lead to higher steady state glucose. We studied published literature for experiments where a stable and sustained increase or decrease in insulin was achieved and then the effect on fasting glucose and GTT studied.

### Fasting versus post feeding glucose in STZ rats

Streptozotocin induced diabetes is a popular model of rodent diabetes (*Akbarzadeh et al., 2007*; *Gajdosík et al., 1999*). STZ acts by specifically destroying the insulin producing β-cells of the pancreatic islets (*Szkudelski, 2001*). A low dose of STZ that destroys a substantial population of β cells but does not lead to total destruction of their population is often perceived as a good model for T2D, whereas a high dose of STZ that destroys the β-cell population almost entirely is perceived as a model of T1D (*Akbarzadeh et al., 2007*; *Asrafuzzaman et al., 2017*; *Freitas et al., 2015*; *Gajdosík et al., 1999*; *Kagami et al., 2008*; *Kovacs et al., 2011*; *Szkudelski, 2001*; *Zhang et al., 2008*). We searched literature to look for studies that carefully differentiated between steady state glucose from post load glucose in STZ models but did not find any studies that make this distinction clear. Therefore, we designed and conducted experiments to differentially study the steady state and perturbed state glucose levels in rats treated with STZ.

### Theoretical and mathematical considerations

In this approach we elaborate on the theoretical underpinnings of insulin-glucose relationship. We also explore possible explanations for the unexpectedly consistent failure of experimental insulin signal impairment to alter steady state glucose level. Simultaneously we make differential predictions from alternative homeostasis models that can be tested in human epidemiological data. The two models which we use here answer the critical question in glucose homeostasis: whether the fasting/steady state glucose level is a consequential balance between glucose production and glucose utilisation rates (consequential steady state CSS) or whether there is a target glucose level that is maintained by sensing and correcting any changes in it (targeted steady state TSS).

*Analysis of insulin-glucose relationship in steady and perturbed state in human data: Epidemiological inquiry*

Here we use human epidemiological data to test the correlational predictions made by the CSS vs. TSS models of glucose homeostasis.

# MATERIALS AND METHODS

We will outline the methods used in each of the five approaches described in the last section of the introduction.

## Systematic review registration

1. The four meta-analyses were registered in PROSPERO.
2. We did not start the project with the intention of doing these systematic reviews. Our primary goal was to test the alternative evolutionary hypotheses for the origins of type 2 diabetes (*Watve & Diwekar-Joshi, 2016*). While pursuing this work we noticed certain basic anomalies in the classical theories of diabetes. At the stage when we thought of the question addressed in this paper, we had already scanned much of the literature used here.
3. To ensure that our review is unbiased and rigorous enough, we repeated the literature search following PRISMA and PROSPERO guidelines. Therefore, much of the search for the meta-analyses was done before the actual PROSPERO registration. The secondary screening, data extraction and the data analysis was done during the time required after the submission to the PROSPERO and the actual approval of the registration by them.
4. But we were glad to find after following proper procedures that our prior work met the standards of unbiased and rigorous review.
5. Details about the search strategies, quality assessment and data analysis for the four meta-analyses are included in Supplemental Information 1.
6. The flowcharts for the search strategies and screening procedures followed are given in the form of PRISMA flowcharts and checklists and are given as Supplementary Information files.

## Systematic review/meta-analyses: search and screening

1. Manawa Diwekar-Joshi and Milind Watve decided the search strategy and the actual searches and screening based on the strategy was done by Manawa Diwekar-Joshi.
2. Data analysis was done by both the authors.
3. Disagreements were dissolved by discussion and input from other lab members.
4. The referees for the manuscript were other post-doc and graduate students of the Watve Lab.

**Table 1** IRKO meta-analysis: Systematic literature review for studies on tissue specific insulin receptor knock-outs.

| Meta-analysis→<br>Task performed ↓ | Insulin receptor knockout |
| --- | --- |
| Key word(s) used for the first search on the PubMed/ MEDLINE data-base | "Insulin receptor knockout" |
| Number of hits in the first search | 78 |
| Inclusion criteria for primary screening | Study showing experiments with IRKOs in which GTT curve has been reported |
| Number of papers shortlisted based on primary screening | 36 |
| Inclusion criteria for secondary screening | Study showing similar methods of making the insulin receptor knockout; had fasting and post glucose bolus readings of the control and knockout |
| Number of papers shortlisted based on screening the full-text and back referencing (data extracted from these papers) | 16 |
| List of publications used in the final analysis: references given here | *Blüher et al. (2002)*, *Brüning et al. (1998)*, *Cohen et al. (2004)*, *Dodson Michael et al. (2000)*, *Ealey et al. (2008)*, *Escribano et al. (2009)*, *Guerra et al. (2001)*, *Haas et al. (2012)*, *Kawamori et al. (2009)*, *Lauro et al. (1998)*, *Mauvais-Jarvis et al. (2000)*, *Okada et al. (2007)*, *Otani (2003)*, *Sakaguchi et al. (2017)*, *Softic et al. (2016)*, *Wojtaszewski et al. (1999)* |

## Statistical approach (used for all the meta-analyses)

Although we short-listed articles that used similar methods, small differences in protocols can make considerable differences in the results. There is substantial variation in results across studies. Therefore, we use non-parametric methods for analysing the pooled data. We first look at in how many of the experiments the treatment means are greater than the control means and in how many they are smaller. If this difference is significant, we conclude that there is enough qualitative consistency across experiments to reach a reliable inference. If there is a consistent direction of difference, we look at how many are individually significant. As a conservative approach we avoid pooling data quantitatively since across studies there are differences in age or weight of animals, number of days after treatment, number of hours of fasting and other variables. This approach is maintained for the analysis of all the meta-analyses.

## Systematic review of experiments involving tissue specific insulin receptor knock-outs

The details of the method of the systematic literature review involving the tissue specific insulin receptor knock-outs are given in Table 1. The systematic review was registered in PROSPERO (ID: CRD42019132379).The details of the experiments of the shortlisted studies can be seen in Table 1 of the Supplemental Information 1 which shows that similar methods have been utilised to create the knockouts and therefore a comparative analysis is justified.

## Systematic review of experiments to chronically raise or lower insulin levels

### *Increase in insulin*

A model for sustained increase in insulin levels is a knock-out or inhibition of the insulin degrading enzyme (IDE). An interplay between insulin secretion and insulin degradation

**Table 2 IDE inhibition meta-analysis: Systematic literature review for studies on insulin degrading enzyme inhibition/knock-out.**

| Meta-analysis→<br>Task performed ↓ | Insulin degrading enzyme |
|---|---|
| Key word used for the first search on the PubMed/MEDLINE database | "Insulin degrading enzyme" |
| Number of hits in the first search | 1179 |
| Inclusion criteria for primary screening | Studies showing experiments with inhibition of IDE and GTT |
| Number of papers shortlisted based on primary screening | 33 |
| Inclusion criteria for secondary screening | Study showed experiments with IDE inhibition, had fasting and post-glucose load readings of control and IDE inhibition |
| Number of papers shortlisted based on screening the full-text and back referencing (data extracted from these papers) | 6 |
| List of the publications used in the final analysis (all were rodent studies; references given here) | *Abdul-Hay et al. (2011)*, *Deprez-Poulain et al. (2015)*, *Durham et al. (2015)*, *Farris et al. (2003)*, *Maianti et al. (2014)*, *Villa-Pérez et al. (2018)* |

maintains the level of insulin in plasma (*Authier, Posner & Bergeron, 1996*; *Duckworth, Bennett & Hamel, 1998*; *Hulse, Ralat & Wei-Jen, 2009*; *Shen et al., 2006*). Plasma insulin has a half-life of 4–9 min (*Hulse, Ralat & Wei-Jen, 2009*; *Tomasi et al., 1967*) and it is degraded predominantly by the insulin degrading enzyme (IDE) (*Hulse, Ralat & Wei-Jen, 2009*; *Shen et al., 2006*). Inhibition of IDE has been considered as a therapeutic option for type 2 diabetes with limited and questionable success (*Costes & Butler, 2014*; *Maianti et al., 2014*). We performed a systematic literature review to find out experiments in which IDE was inhibited to obtain a sustained high plasma insulin level and, in such animals, GTT was performed (Table 2) (PROSPERO registration ID: CRD42019140619).

### Decrease in insulin

We performed a systematic literature review for experiments in which there was sustained suppression of insulin production. Two insulin suppressing agents have been repeatedly used to lower insulin production in rodent models as well as in humans.

(i) Diazoxide (DZX): Diazoxide is a potassium channel activator which causes reduction in insulin secretion by the β-cells by keeping the cells in a hyperpolarized state by opening the channel (*Panten et al., 1989*). It has been used as a drug to modulate insulin secretion for research and therapeutic purposes (*Doyle, 2003*).

(ii) Octreotide (OCT): Octreotide is a somatostatin analogue which inhibits insulin and growth hormone. It has been used to reduce insulin secretion in vitro and in vivo (*Lamberts et al., 1996*).

We searched the literature systematically for studies where the insulin levels have been altered using either DZX or OCT and glucose tolerance has been examined using a GTT after DZX/OCT treatment (Table 3) (PROSPERO registration ID for DZX: CRD42020141688; PROSPERO registration ID for OCT: CRD42020141464). It should be noted that this literature includes a significant proportion of human trials. We also searched literature for studies in which insulin was suppressed by other methods.

**Table 3 Diazoxide and Octreotide meta-analyses: Systematic literature review for studies on insulin suppression with diazoxide and octreotide.**

| Meta-analyses→ Task performed ↓ | Diazoxide | Octreotide |
|---|---|---|
| Key word used for the first search on the PubMed/MEDLINE data-base | "Diazoxide and diabetes"; "insulin suppression" | "Octreotide and diabetes"; "insulin suppression |
| Number of hits in the first search | 1,043 | 1,202 |
| Inclusion criteria for primary screening | Study shows stable insulin suppression using diazoxide and a GTT has been performed after insulin suppression. | Study shows stable insulin suppression using octreotide and a GTT has been performed after insulin suppression. |
| Papers shortlisted based on primary screening | 239 | 289 |
| Inclusion criteria for secondary screening | Study showed similarities in the concentration of diazoxide used; and had fasting and post glucose bolus readings of the control and diazoxide subjects | Study showed similarities in the concentration of octreotide used; and had fasting and post glucose bolus readings of the control and octreotide subjects |
| Papers shortlisted based on screening the full-text and back referencing (data extracted from these papers) | Rodent studies (2) Human studies (6) | Rodent studies (0) Human studies (10) |
| List of human studies used in the final analysis | *Brauner et al. (2016)*, *Due et al. (2007)*, *Ramanathan, Arbeláez & Cryer (2011)*, *Schreuder et al. (2005)*, *Van Boekel et al. (2008)*, *Wigand & Blackard (1979)* | *Breckenridge et al. (2007)*, *Candrina, Gussago & Giustina (1988)*, *Davies et al. (1986)*, *Giustina et al. (1991)*, *Johnston et al. (1986)*, *Madsen et al. (2011)*, *Parkinson et al. (2002)*, *Ronchi et al. (2002)*, *Williams et al. (1986, 1988)* |
| List of rodent studies used in the final analysis | *Leahy, Bumbalo & Chen (1994)*, *Matsuda et al. (2002)* | None |

## Steady state versus perturbed state glucose in STZ rats
### Animal model and conditions
*Ethics approval*

The experiments performed on Sprague Dawley (SD) rats had been approved by the Institutional Animal Ethics Committee at IISER, Pune (Protocol Number IISER/IAEC/2016-02/006) constituted by the Committee for the Purpose of Control and Supervision of Experiments on Animals (CPCSEA), Govt. of India.

*Housing of the animals*

All the rats were housed in a facility with a temperature of 23 ± 2 °C and a 12-h light/dark cycle with standard rat chow (Altromin rat/mice maintenance diet) and water available ad libitum. The bedding of the cages was changed every three days and every day after injection of STZ. There were no extra measures taken for the enrichment of the animals.

*Euthanasia*

The animals were not euthanized before the end of the experiments. At the end of the 12 days of glucose and insulin readings, the animals were euthanized with an intraperitoneal injection of thiopentone (100–120 mg/kg body weight).

### STZ treatment for insulin suppression

Male SD rats weighing 180–200 g were injected with STZ at 50 mg/kg body weight. The STZ was dissolved in Citrate Buffer (Citric Acid: 0.1M and Sodium Citrate: 0.1M). Injection of Citrate Buffer alone was used as control.

### Fasting and post-feeding glucose in 12 day follow up

Three days after the STZ injection, the rats were fasted for 16 h and glucose was measured using the hand held Accu-Chek Glucometer. The rats were then given 40 g of Standard Chow for 8 h. Food was weighed and post-feeding glucose was measured after three hours. The protocol was repeated for 12 days and body weight, food weight and glucose readings were taken daily. A total of 12 animals per group were used for this experiment.

### Duration of fasting

An experiment was also performed to see how much time was required to reach a steady state of glucose after removal of food. The food was removed from the STZ and Control animals after ad libitum availability and glucose readings were taken after 3, 6, 9, 12 and 16 h. After a recovery of three days, glucose levels were measured only at 16 hours after removing the food. 9 STZ treated animals and 10 Control animals (injected with Citrate Buffer) were used for this experiment.

## Theoretical and mathematical considerations

### Choice of models for glucose homeostasis

The fasting state has been generally accepted to be a steady state for glucose concentration for several reasons. In a given healthy individual the fasting glucose levels are stable in time (*Halter et al., 1985*; *Lerner & Porte, 1972*). The post-meal peak of glucose and insulin returns to the fasting level within a few hours and remains stable over a long time. The fasting state is considered and modelled as a steady state by the widely used HOMA model (*Matthews et al., 1985*; *Turner et al., 1979*). Classically the negative feedback loops are assumed to work through insulin and insulin is taken as a determinant of steady state glucose level. Most popular models of glucose homeostasis work on this assumption although non-steady state models of insulin resistance exist (*Palumbo et al., 2013*).

### Models used for this analysis

A critical question in glucose homeostasis is whether the fasting steady state glucose level is a consequential balance between glucose production and glucose utilisation rates (consequential steady state CSS) or whether there is a target glucose level that is maintained by sensing and correcting any changes in it (targeted steady state TSS). The difference in the two can be visualised by the tank water level analogy (Fig. 2). If a tank has an input tap releasing water in it at a constant rate and has an outlet at the bottom through which water escapes proportionate to the pressure of the water column, a steady state is invariably reached (Fig. 2A). The steady state level is decided by the rate of intake and the size of the outlet. This is a CSS which will change with any change in

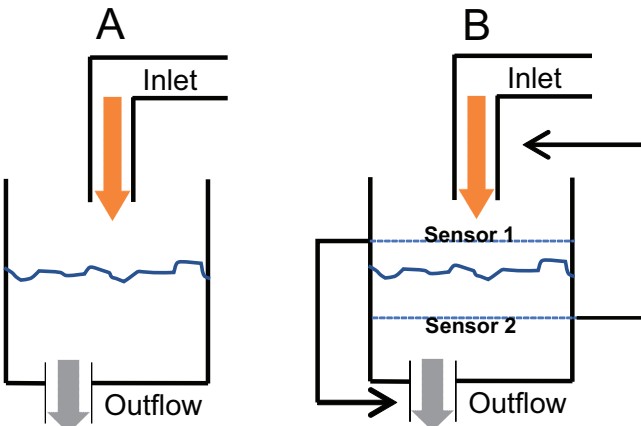

**Figure 2 Consequential steady state (CSS) and targeted steady state (TSS) models of glucose homeostasis.** The CSS (A) and TSS (B) models of homeostasis illustrated with a tank water level analogy. In CSS a change in the size of inlet or outlet tap, analogous to insulin sensitivity can change the steady state level. In a TSS model, a change in the tap size will alter the time required to reach a steady state but will not change the steady state level.

the size/capacity of the input or the outlet tap. In contrast to CSS, in a TSS there is a desired water level and sensors are placed above and below the desired level such that when the level goes below the lower sensor the input is switched on or its rate increased and/or output switched off or its rate decreased (Fig. 2B). Refer to Supplementary Information 2 for the details of the CSS model.

## Analysis of insulin-glucose relationship in steady and perturbed state in human data: Epidemiological inquiry

### Epidemiological data

The three data sets used here come from two different studies: (i) Coronary Risk of Insulin Sensitivity in Indian Subjects (CRISIS) study, Pune, India (*Yajnik et al., 2007*) and (ii) Newcastle Heart Project (NHP), UK (*Bhopal et al., 1999*). Data from the latter is divided into two groups as the subjects belong to different ethnicities namely European white and south Asian and we will prefer to analyse the two groups separately since certain ethnic differences are likely to be present in the tendency to develop metabolic syndrome (*Bhopal, 2013*; *Gujral et al., 2013*). Hence all the comparison of predictions with the data has been done independently for the three data sets. All the studies are population surveys that include non-diabetic (fasting glucose values less than 110 mg/dl) and diabetic individuals (fasting glucose values above 110 mg/dl) and the clinical history, morphometric parameters, glucose and insulin during fasting and oral glucose tolerance test (OGTT) of the subjects were recorded. In the analysis, we included only the non-diabetic groups in which the homeostatic mechanism can be assumed to be intact and therefore any hypothesis about it can be tested. Most of the individuals in the diabetic group would be under different drug regime affecting glucose-insulin dynamics in different ways and therefore we exclude that group for the analysis.

### Statistics

Linear regression and correlation were used to compare the glucose-insulin relationship in steady state (fasting) versus perturbed state (post-glucose load) in the three data sets along with the relationships between HOMA-IR and HOMA-β derived from the fasting data.

## RESULTS

### Systematic review of experiments involving tissue specific insulin receptor knock-outs

#### Pooled results of IRKO experiments

We shortlisted 16 articles describing 46 independent experiments in which glucose tolerance curves of insulin receptor knock-outs and controls were compared (Table 1 of Supplemental Information 1). The experiments could be segregated in four different tissue specific knockouts for the analysis: fat/adipose insulin receptor knock-out or FIRKO (Fig. 3), Muscle insulin receptor knock-out or MIRKO (Fig. 4), liver insulin receptor knock-out or LIRKO (Fig. 5) and β-cell insulin receptor knock-out or βIRKO (Fig. 6). A generalised trend in the total picture summed up over all four IRKOs was that along the GTT curve, significantly higher glucose levels are seen in the knockouts as compared to the controls, particularly and consistently at 30, 60 and 120 min. However, the fasting glucose level was not significantly different. In some studies, fasting glucose was significantly higher in the knock-outs than the controls, however in some other studies it was significantly lower as well. In 29 out of 46 experiments there was no significant difference (Table 4) in the fasting glucose levels of knock-outs and controls. This trend was consistently seen in MIRKO (Fig. 4), LIRKO (Fig. 5) and βIRKO (Fig. 6). Only in FIRKO (Fig. 3) there were greater number of studies showing fasting glucose significantly higher in the knock-outs than in the controls(*Guerra et al., 2001*; *Sakaguchi et al., 2017*; *Softic et al., 2016*), but in the non-parametric analysis the collective trend was not significant Table 4. These inconsistencies in the FIRKO fasting glucose levels compared to the MIRKO and LIRKO could be explained with the help of the duration of fasting used for the glucose tolerance tests. Although the glucose levels in the wild type are higher than the FIRKO/BATIRKO in the fasting conditions, there could be possible reasons for that. For example, in the study (*Sakaguchi et al., 2017*), the fasting for the glucose tolerance test was carried out only for 6 hours as compared to 16 hours/overnight in other studies. Secondly, one of the knock-outs used in this study is a double knock-out of insulin receptor as well as the insulin-like growth factor 1 receptor which could be a possible reason for the higher glucose levels in the fasting condition. In case of the *Guerra et al., 2001*; *Softic et al., 2016*, the animals have been fasted ON for the glucose tolerance test and the tests have been performed on FIRKO and WT of different ages. The impairment of glucose tolerance increases with age, though this also is not seen consistently across all the studies. In the case of *Blüher et al. (2002)*, the fasting duration for the GTT is 16 h, highest in all the studies and in this case the treated glucose levels in the fasting condition are equal to or lower than that of the controls. It is important to note that in FRIKO,

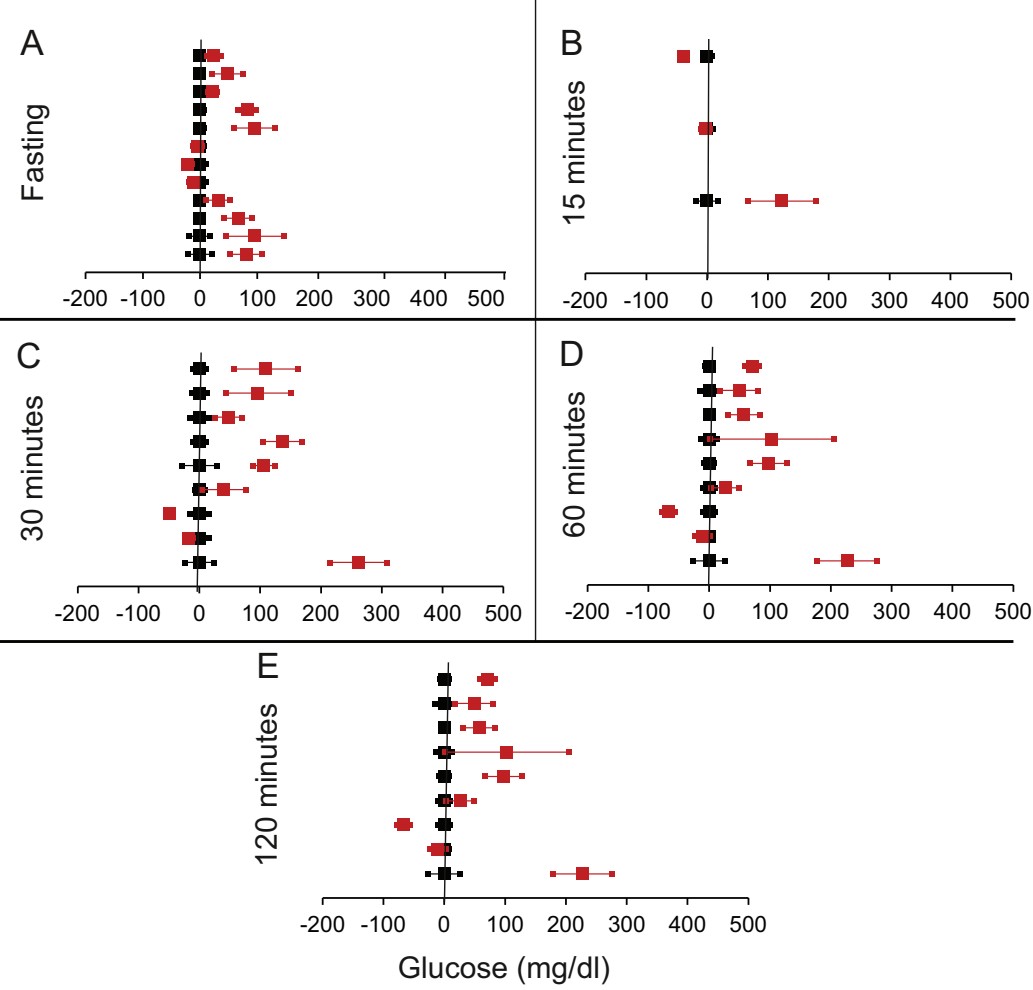

**Figure 3 Steady state and perturbed state glucose for control and FIRKO.** Glucose levels for control (black squares) and **FIRKO** (red squares) at steady state and perturbed state. The *X*-axis represents the glucose levels and the *Y*-axis represents experiments from different studies from the shortlisted papers in which control and FIRKO were compared using an OGTT.FIRKO glucose levels are normalised to that of the control and the difference is expressed with ±95% CI. Glucose levels of control are expressed as 0 ± 95% CI. Steady state is represented by the fasting glucose (A) and the perturbed state is represented by different time points post-glucose load when the readings are taken: (B) 15 min (C) 30 min (D) 60 min and (E) 120 min.

although there are maximum number of cases where in the treated is higher than the control in the fasting condition, this trend is not significant in the non-parametric analysis (Table 4). Also, only in FIRKO, the 30, 60- and 120-min glucose was not significantly different in the knock-outs than the controls (Fig. 3). It is notable that in none of LIRKO experiments the fasting sugar was significantly higher than the controls. This contradicts the classical belief that liver insulin resistance is mainly responsible for fasting hyperglycaemia in T2D (*Bock et al., 2007*; *Johnson et al., 1972*).

A possible problem in comparing fasting glucose across different studies was that different fasting intervals have been used ranging from 4 to 16 h. No study clearly reported

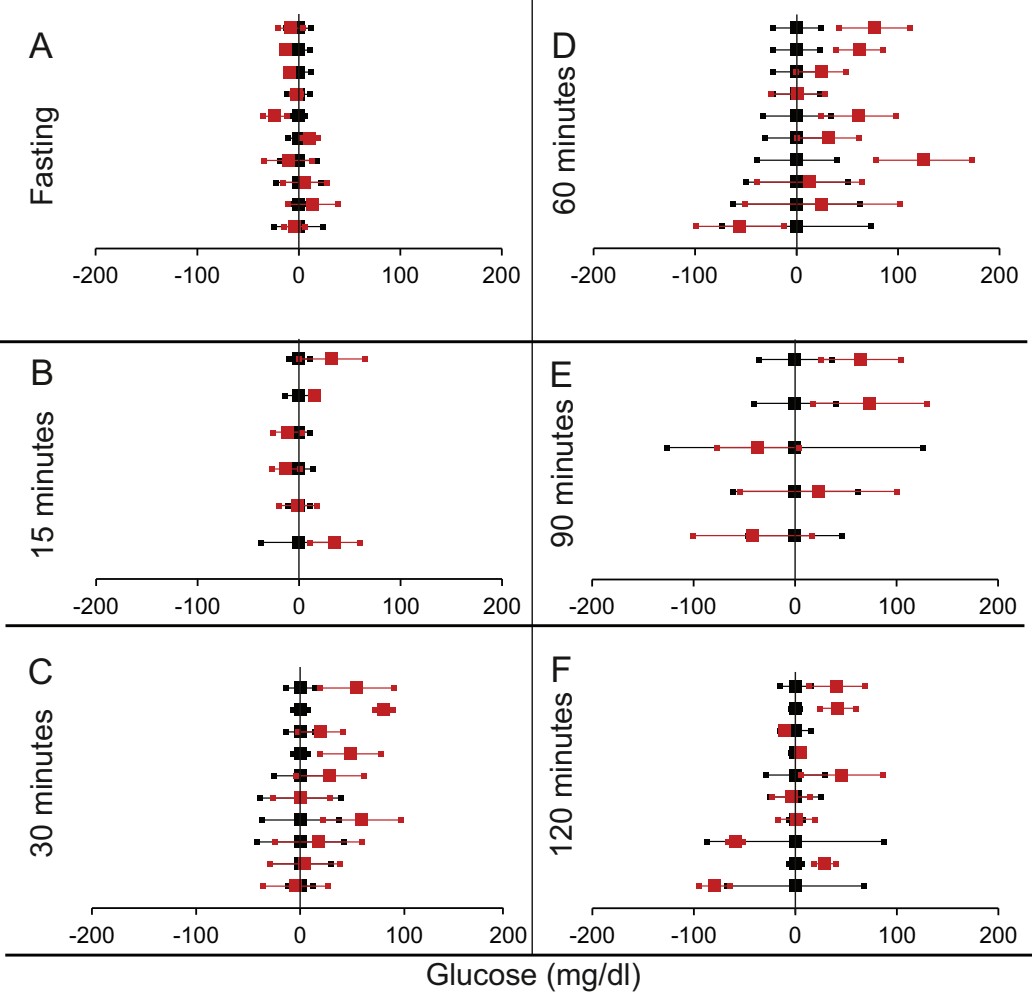

**Figure 4  Steady state and perturbed state glucose for control and MIRKO.** Glucose levels for control (black squares) and **MIRKO** (red squares) at steady state and perturbed state. The *X*-axis represents the glucose levels and the *Y*-axis represents experiments from different studies from the shortlisted papers in which control and MIRKO were compared using an OGTT.MIRKO glucose levels are normalised to that of the control and the difference is expressed with ±95% CI. Glucose levels of control are expressed as 0 ± 95% CI. Steady state is represented by the fasting glucose (A) and the perturbed state is represented by different time points post-glucose load when the readings are taken: (B) 15 min (C) 30 min (D) 60 min (E) 90 min and (F) 120 min. Note that fasting glucose does not differ from the control in any of the experiments.               

how much time is required to reach a steady state in a knock-out. In 10 of the experiments in which fasting time was reported as 16 h, none had fasting sugar significantly different from controls. In the 13 experiments in which it was high, the fasting duration was between 4 and 12 h or not precisely reported. Therefore, it is likely that in at least some of the experiments, glucose steady state was not yet achieved at the time point defined as fasting. This bias increases the probability that higher fasting glucose is reported for the knock-outs. However, since we do not see a significant difference in the collective analysis, the inference that IRKO does not alter fasting glucose is unlikely to

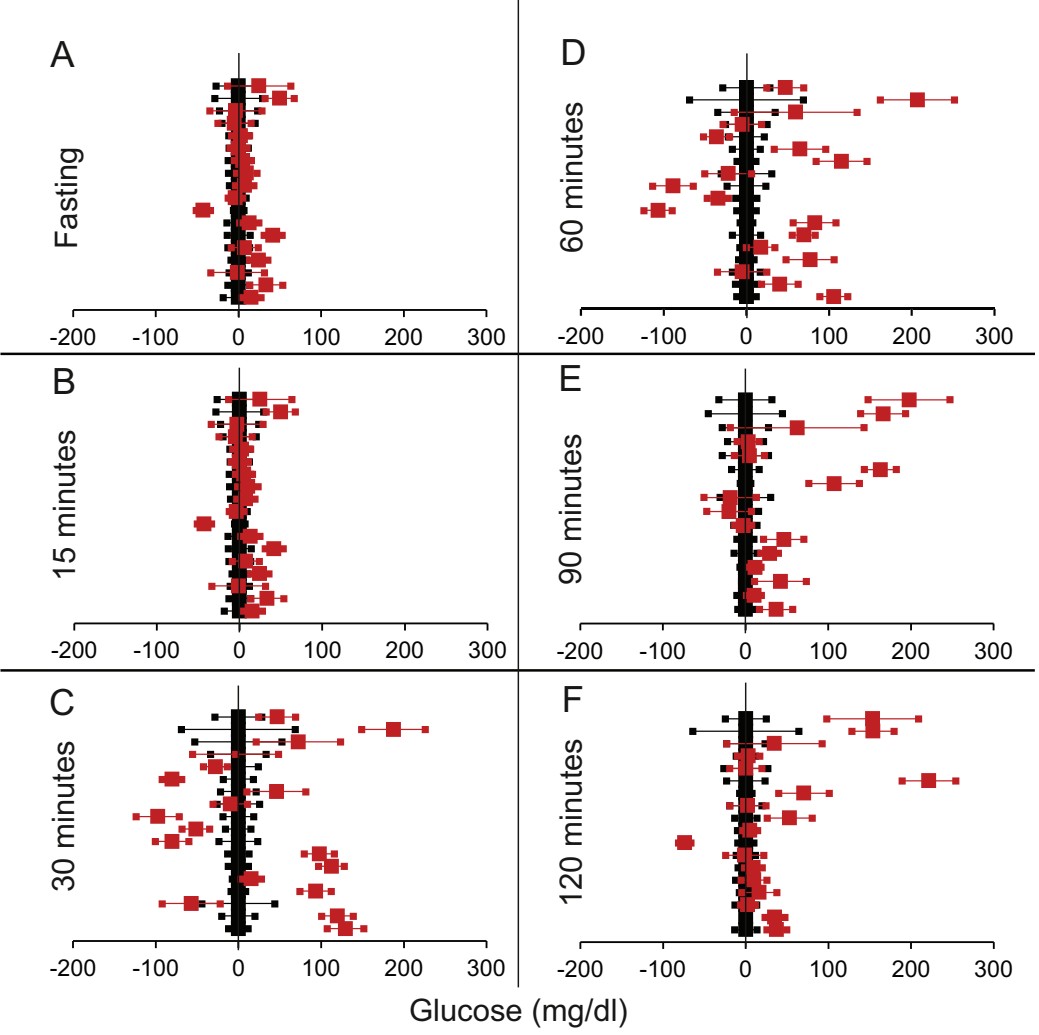

**Figure 5 Steady state and perturbed state glucose for control and LIRKO.** Glucose levels for control (black squares) and **LIRKO** (red squares) at steady state and perturbed state. The *X*-axis represents the glucose levels and the *Y*-axis represents experiments from different studies from the shortlisted papers in which control and LIRKO were compared using an OGTT.LIRKO glucose levels are normalised to that of the control and the difference is expressed with ±95% CI. Glucose levels of control are expressed as 0 ± 95% CI. Steady state is represented by the fasting glucose (A) and the perturbed state is represented by different time points post-glucose load when the readings are taken: (B) 15 min (C) 30 min (D) 60 min (E) 90 min and (F) 120 min. Note that inconsistent with classical belief, liver specific insulin receptor knock-out does not show significant effect on fasting glucose in any of the experiments. On the other hand, post load glucose is consistently higher.

be a result of the bias. In fact, any possible correction to the bias might further reduce the apparent residual difference. Therefore, in spite of some inconsistency across studies, a robust generalisation is that IRKOs have significantly increased plasma glucose over controls at 30–120 min post-glucose load but they do not appear to affect steady state fasting glucose. The time required to reach the steady state is evidently increased.

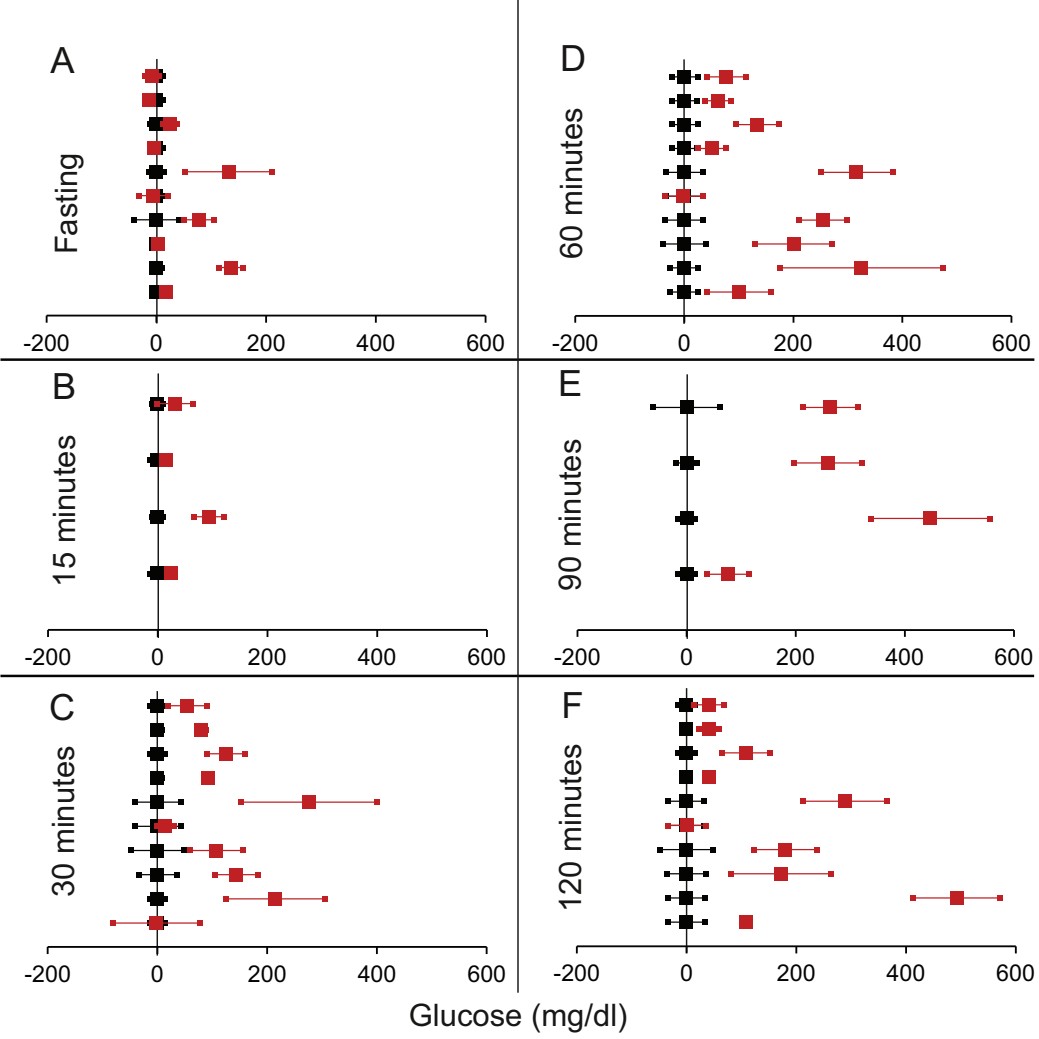

**Figure 6 Steady state and perturbed state glucose for control and βIRKO.** Glucose levels for control (black squares) and **βIRKO** (red squares) at steady state and perturbed state. The *X*-axis represents the glucose levels and the *Y*-axis represents experiments from different studies from the shortlisted papers in which control and βIRKO were compared using an OGTT.βIRKO glucose levels are normalised to that of the control and the difference is expressed with ±95% CI. Glucose levels of control are expressed as 0 ± 95% CI. Steady state is represented by the fasting glucose (A) and the perturbed state is represented by different time points post-glucose load when the readings are taken: (B) 15 min (C) 30 min (D) 60 min (E) 90 min and (F) 120 min. Note that fasting glucose does not differ from the control in any of the experiments.

## Systematic review of experiments to chronically raise or lower insulin levels

### *Increase in insulin by suppression of IDE*

We found six publications that described 18 experiments that allowed comparison of GTT between raised insulin groups and control group (Table 2 of Supplemental Information 1). Analysis revealed no significant difference in the fasting glucose. In only one out of 18 experiments the treatment group had lower fasting glucose than the control. During the GTT curve, at 90 and 120 min the difference between treatment and control were

**Table 4 Analysis of IRKO meta-analysis.** Collective analysis of the fasting and post-feeding glucose levels in the control and IRKOs: The table shows, out of the total number of experiments used for the analysis, in how many the mean of the knockouts (T) was greater than the control means (C) and in how many the trend was reverse. This relative position of the means across studies is compared non-parametrically to see whether the trend across studies was non-random, significant ones being indicated by asterisk. The table also gives in how many studies T was significantly greater than C and vice versa. It can be seen that for fasting glucose the difference is not significant in majority of studies and where there is statistical significance there is lack of consistency across studies. However, at 30, 60 and 120 min the knockouts have consistently elevated levels of glucose as compared to the corresponding controls.

| | Total studies | T > C | T < C | p Using chi square | T > C Individually significant | T < C Individually significant |
|---|---|---|---|---|---|---|
| All IRKOs | | | | | | |
| Fasting | 46 | 25 | 20 | 0.454 | 13 | 4 |
| 15 min | 14 | 7 | 7 | 0.999 | 4 | 2 |
| 30 min | 40 | 36 | 4 | <0.0001* | 22 | 1 |
| 60 min | 40 | 36 | 4 | <0.0001* | 24 | 1 |
| 120 min | 46 | 37 | 9 | <0.0001* | 24 | 2 |
| FIRKO | | | | | | |
| Fasting | 12 | 9 | 3 | 0.083 | 9 | 1 |
| 15 min | 3 | 1 | 2 | 0.566 | 1 | 1 |
| 30 min | 9 | 7 | 2 | 0.095 | 6 | 1 |
| 60 min | 9 | 7 | 2 | 0.095 | 5 | 1 |
| 120 min | 12 | 9 | 3 | 0.83 | 7 | 2 |
| MIRKO | | | | | | |
| Fasting | 10 | 3 | 7 | 0.205 | 0 | 2 |
| 15 min | 6 | 3 | 3 | 0.999 | 1 | 0 |
| 30 min | 10 | 9 | 1 | 0.011* | 3 | 0 |
| 60 min | 10 | 9 | 1 | 0.011* | 3 | 0 |
| 120 min | 10 | 6 | 4 | 0.527 | 3 | 0 |
| LIRKO | | | | | | |
| Fasting | 9 | 4 | 5 | 0.739 | 0 | 0 |
| 15 min | 1 | 1 | 0 | N.A. | 0 | 0 |
| 30 min | 9 | 9 | 0 | 0.003* | 6 | 0 |
| 60 min | 9 | 9 | 0 | 0.003* | 7 | 0 |
| 90 min | 9 | 9 | 0 | 0.003* | 5 | 0 |
| 120 min | 9 | 7 | 2 | 0.094 | 4 | 0 |
| βIRKO | | | | | | |
| Fasting | 8 | 6 | 2 | 0.157 | 4 | 0 |
| 15 min | 2 | 2 | 0 | 0.157 | 2 | 0 |
| 30 min | 8 | 7 | 1 | 0.033* | 6 | 0 |
| 60 min | 8 | 7 | 1 | 0.033* | 7 | 0 |
| 90 min | 4 | 4 | 0 | 0.046* | 4 | 0 |
| 120 min | 8 | 8 | 0 | 0.0046* | 7 | 0 |

significant but in the opposite direction of the expectation (Table 5; Fig. 7). While rise in insulin level should reduce plasma glucose, it increased in 15 out of 18 studies, two of which were individually significant and the difference was significant in non-parametric

**Table 5 Analysis of IDE inhibition meta-analysis.** Analysis of the fasting and post-feeding glucose levels in the control and **IDE-inhibition:** the table shows, out of the total number of experiments used for the analysis, in how many the mean of the IDE inhibition models (T) was greater than the control means (C) and in how many the trend was reverse. This relative position of the means across studies is compared non-parametrically to see whether the trend across studies was non-random, significant ones being indicated by asterisk. The table also gives in how many studies T was significantly greater than C and vice versa. There is no significant difference in the control at treated fasting glucose level. At 90 and 120 min the difference between the control and treated was significant, but in the opposite direction.

| | Total experiments | T > C | T < C | p Using chi square | T > C Individually significant | T < C Individually significant |
|---|---|---|---|---|---|---|
| Fasting | 18 | 12 | 6 | 0.157 | 0 | 1 |
| 15 minutes | 14 | 7 | 7 | 0.999 | 3 | 5 |
| 30 minutes | 18 | 10 | 8 | 0.637 | 5 | 2 |
| 60 minutes | 18 | 11 | 7 | 0.346 | 5 | 1 |
| 90 minutes | 16 | 13 | 3 | 0.012* | 4 | 0 |
| 120 minutes | 18 | 15 | 2 | 0.002* | 2 | 1 |

collective analysis (Table 5; Fig. 7). Across all time points along the GTT, the plasma glucose in the treated group was greater than the control group in majority of the experiments. Thus, in this class of experiments increasing insulin failed to reduce glucose at the steady state as well as post glucose load.

### Decrease in insulin: suppression by diazoxide or octreotide

We found eight articles describing 14 experiments for diazoxide treatment and 10 articles with 15 experiments for octreotide treatment (Tables 3 and 4 from Supplemental Information 1 respectively). It can be seen from Table 6 and Fig. 8 that for both insulin suppressing agents, suppression of insulin did not result into increased fasting glucose. Further at 120 min post glucose load there was a marginally significant rise in glucose in the insulin suppressed group as compared to control group. This demonstrates that pharmacological suppression of insulin was unable to raise plasma glucose level in a fasting steady state. There was somewhat inconsistent but significant rise post glucose load. Diazoxide is the only agent in which both rodent and human data are available. The results of the two are very similar and separating them does not alter the significance.

We found more means of insulin suppression in which GTT after suppression was reported, but there were not many published replications of the experiments coming independently from different research groups. Therefore, any systematic review was not warranted. We briefly review their results here.

### Suppression by protein restriction

Dietary protein deprivation is another method of insulin suppression. This also led to a decrease in plasma insulin levels; however fasting glucose levels did not increase (*Schteingart et al., 1979*).

### Suppression by insulin siRNA

Transgenic mice for insulin-siRNA along with IDE overexpression, showed decreased levels of insulin. Again the fasting glucose levels remained normal while there was

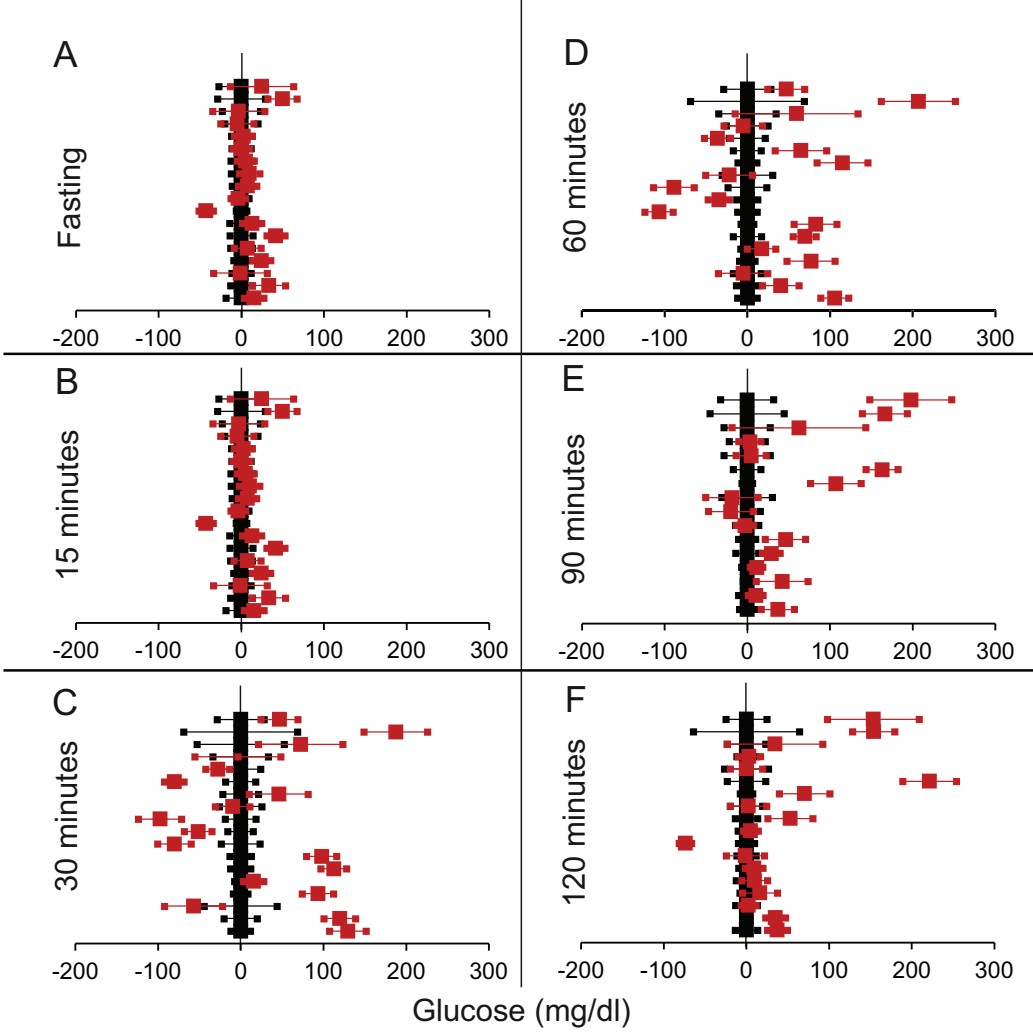

**Figure 7 Steady state and perturbed state glucose for control and IDE inhibition.** Glucose levels for control (black squares) and **IDE-inhibition** (red squares) models at steady and perturbed state. The *X*-axis represents the glucose levels and the *Y*-axis represents experiments from different studies from the shortlisted papers in which control and IDE-inhibition were compared using an OGTT. IDE-inhibition glucose levels are normalised to that of the control and the difference is expressed with ±95% CI. Glucose levels of control are expressed as 0 ± 95% CI. Steady state is represented by the fasting glucose (A) and the perturbed state is represented by different time points (B–F) post-glucose load when the readings are taken: (B) 15 min (C) 30 min (D) 60 min (E) 90 min and (F) 120 min. Note that fasting glucose does not differ significantly from the control. At 90 and 120 min the trend is higher mean glucose than control which is contrary to the expectation in an experiment with sustainable rise in insulin.

a change in glucose tolerance curve (Fig. 9) (*Hwang et al., 2007*). The curves in Fig. 9 are typical of insulin receptor knockout or insulin suppression experiments where, in individuals with impaired insulin signalling the glucose peak is higher which returns to steady state much later than the controls, but the fasting steady state level is not different.

**Table 6 Analysis of DZX and OCT meta-analyses.** Analysis of the fasting and post-feeding glucose levels in the control and treatment with Diazoxide or Octreotide: the table shows, out of the total number of experiments used for the analysis, in how many the mean of the treated (T) was greater than the control means (C) and in how many the trend was reverse. This relative position of the means across studies is compared non-parametrically to see whether the trend across studies was non-random, significant ones being indicated by asterisk. The table also gives in how many studies T was significantly greater than C and vice versa.

| | Total studies | T > C | T < C | p Using chi square | T > C Individually significant | T < C Individually Significant |
|---|---|---|---|---|---|---|
| Diazoxide treatment | | | | | | |
| Fasting | 14 | 10 | 3 | 0.052 | 5 | 0 |
| 15 minutes | 7 | 6 | 1 | 0.059 | 1 | 1 |
| 30 minutes | 12 | 10 | 2 | 0.021* | 2 | 1 |
| 60 minutes | 13 | 9 | 4 | 0.166 | 4 | 2 |
| 90 minutes | 3 | 3 | 0 | 0.083 | 2 | 0 |
| 120 minutes | 14 | 10 | 3 | 0.052 | 6 | 1 |
| Octreotide treatment | | | | | | |
| Fasting | 15 | 6 | 7 | 0.781 | 0 | 0 |
| 30 minutes | 14 | 4 | 10 | 0.108 | 0 | 2 |
| 60 minutes | 14 | 4 | 10 | 0.108 | 2 | 1 |
| 90 minutes | 13 | 5 | 8 | 0.405 | 1 | 0 |
| 120 minutes | 15 | 7 | 8 | 0.797 | 1 | 1 |

### Suppression of insulin by partial gene ablation

In rodents, there are two insulin genes *Ins1* and *Ins2* (*Duvillie et al., 1997*). A double knock-out of both the genes results in death, but ablation of either of the genes does not alter the glucose tolerance significantly suggesting redundancy (*Mehran et al., 2012*). There are studies in which one gene is completely knocked out and the other one is a heterozygote (*Dionne et al., 2016*; *Mehran et al., 2012*; *Page et al., 2019*; *Templeman, Clee & Johnson, 2015*; *Templeman et al., 2017*). Reduced insulin gene dosage did not consistently result into fasting hyperglycaemia in these studies although it offered protection against some of the effects of hyperinsulinemia.

### Steady state versus perturbed state glucose in STZ rats

Among the STZ treated rats, all the animals showed significantly higher post load glucose than the control group on all the 12 days sampled. However, in 10 out of the 12 days the 16-h fasting glucose was not significantly different from the control although the variance was substantially greater than that of the control (Fig. 10).

A closer look at the time course of fasting in the two groups revealed that in 4 out of 9 STZ animals the glucose levels reached the normal range but with substantial delay as compared to control animals. In two more animals the levels did not reach the normal range till 16 h but a monotonic decrease continued throughout the period, indicating that their blood glucose may not have reached a steady state in 16 h. Only in three animals the 16-hour glucose was higher than the control range with some indications of stabilising at a higher level. In the time course experiment, the animals are handled

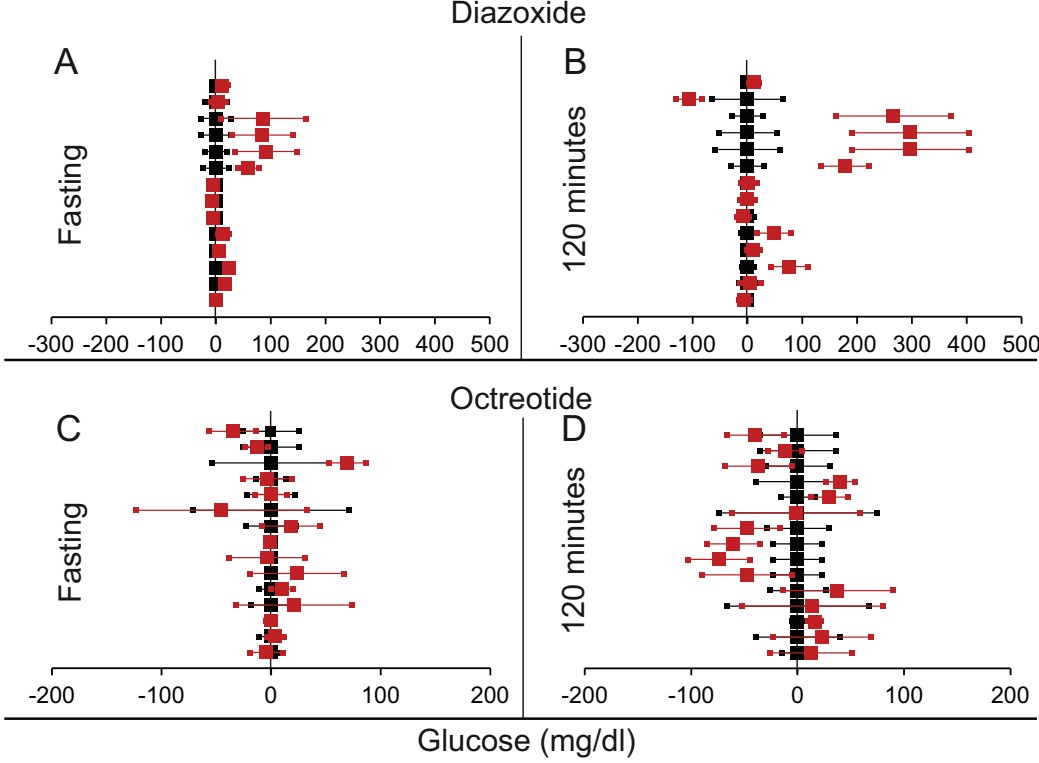

**Figure 8 Steady state and perturbed state glucose for control and insulin suppression.** Glucose levels for control (black squares) and treated (red squares) models at steady and perturbed state. The *X*-axis represents the glucose levels and the *Y*-axis represents experiments from different studies from the shortlisted articles in which control and treated were compared using an OGTT. Glucose levels after treatment with Diazoxide/Octreotide are normalised to that of the control and the difference is expressed with ±95% CI. Glucose levels of control are expressed as 0 ± 95% CI. For Diazoxide treatment, steady state is represented by the fasting glucose (A) and the perturbed state is represented by (B) at 120 min post glucose load. For Octreotide treatment, steady state is represented by the fasting glucose (C) and the perturbed state is represented by (D) at 120 min post-glucose load. Note the inconsistencies across studies.                               

frequently leading to some unavoidable stress, which may have influenced the glucose levels. In the 12 day follow up experiment, plasma glucose is estimated only after 16 hours and here there is no significant difference in the control and STZ animals on 10 out of 12 days. Furthermore, the individuals that showed higher 16 h fasting glucose did not do so consistently. In the 12 days follow up, the distribution of 16 h fasting glucose was typically skewed with one or two outliers having high glucose levels. Interestingly the outliers were not the same animals every day. There was considerable day to day variation in individuals and averaged over the 12 days, none of the STZ animals showed significantly higher fasting glucose than the controls although they consistently showed higher post feeding glucose.

Thus, these experiments show on the one hand that STZ treatment failed to increase steady state glucose levels significantly and consistently. On the other, the STZ animals took substantially longer and rather unpredictable time to reach a steady state and even at 16 h of fasting, all individuals need not have attained a steady state. These results warrant

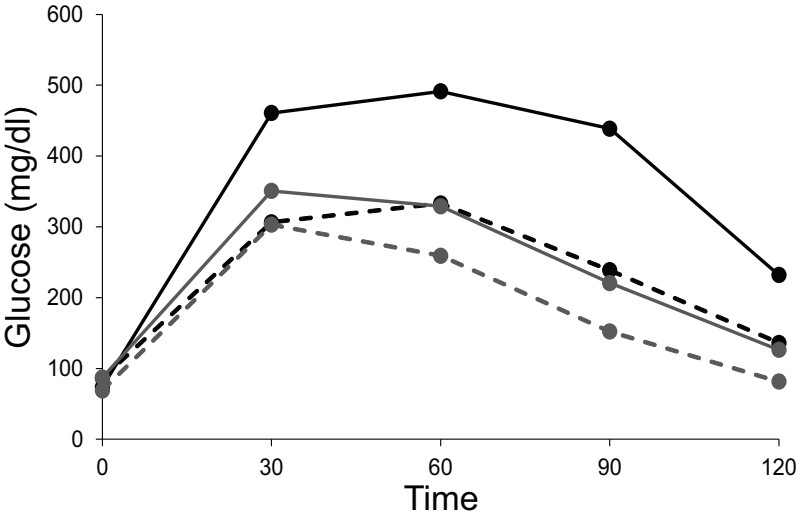

**Figure 9 Steady state and perturbed state glucose after insulin suppression by siRNA.** Intra peritoneal glucose tolerance test of control and siRNA treated mice. Fasting glucose levels in both the siRNA treated (Tg, represented by solid lines) and untreated group (non-Tg, represented by dotted lines) remain unaltered in male (black lines) and female (grey lines) mice. A total of 15 min after the glucose injection, the treated mice show higher glucose levels relative to the untreated mice and this effect is seen throughout till 120 min. Figure redrawn from data by *Hwang et al. (2007)*.

caution against considering fixed hours fasting glucose as steady state glucose in experimental or epidemiological data. While in healthy individuals it is well established that following glucose load a steady state is regained in about two hours, it is possible that in experimental impairment of insulin signalling or in clinical diabetes, plasma glucose takes substantially longer time to reach a steady state and overnight fasting need not represent a steady state in all cases.

## Theoretical and mathematical considerations

In glucose homeostasis, in a fasting state, liver glucose production is analogous to the inlet tap and tissue glucose uptake analogous to the size of the outlet (Fig. 2), both being a function of insulin signalling. Most models of glucose regulation assume CSS (*Bergman, 1989*; *Bergman, 2005*; *Makroglou, Li & Kuang, 2006*; *Matthews et al., 1985*; *Tomasi et al., 1967*; *Turner et al., 1979*; *Palumbo et al., 2013*). According to the CSS, the steady state/ fasting level of glucose (level of water in the water tank analogy) is decided by the difference between the inlet tap (liver glucose production) and outlet (tissue glucose uptake) (Fig. 2A). Both liver glucose production and tissue glucose uptake depend on insulin signalling, hence a change in insulin signalling (change in inlet and outlet flow rates) would result in an altered level of glucose (steady-statewater level would change).

It has not been critically examined whether CSS or TSS describes glucose homeostasis more appropriately. This is important because if TSS model is appropriate, insulin resistance and relative insulin deficiency will not result into altered steady state glucose levels although the time required for reaching a steady state after perturbation might change. Comparing the TSS with the water tank analogy, if insulin resistance affects the

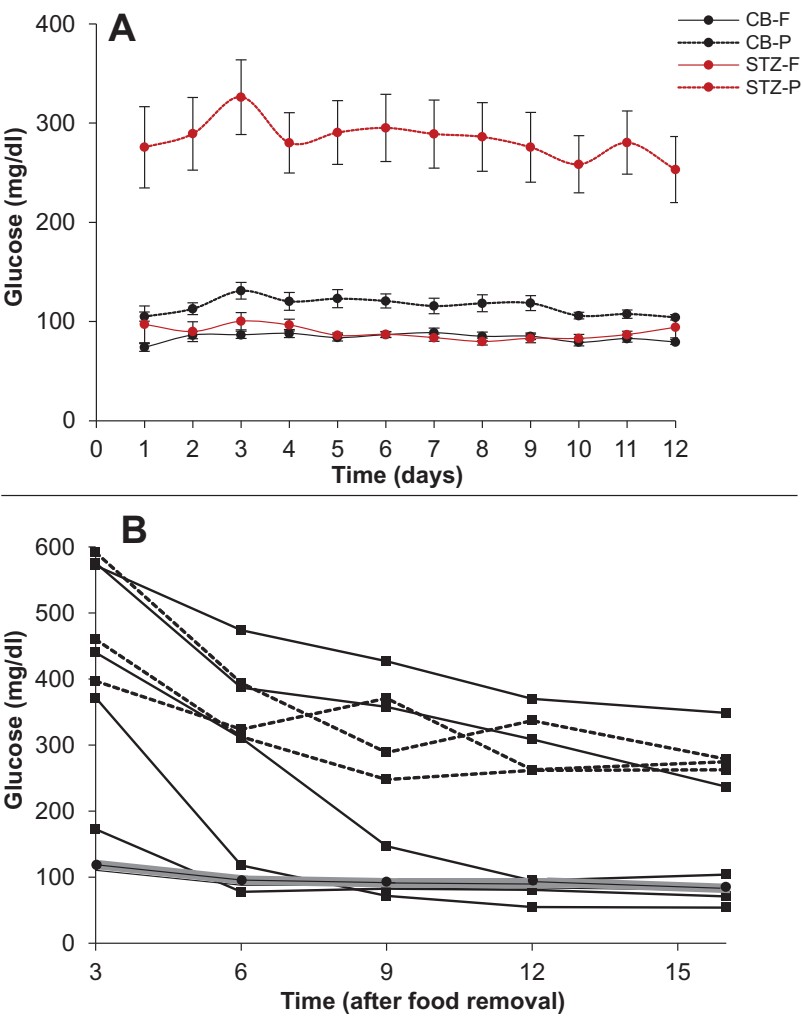

**Figure 10 Steady state and perturbed state glucose after insulin suppression by STZ.** Treatment of SD rats with STZ. (A) The 16 h fasting and post-feeding glucose values of treated (STZ 50mg/kg) and control rats (Citrate buffer CB) over 12 days. $N = 12$ for each group. Note that on 10 out of 12 days the mean fasting glucose of the treated group (STZ-F) was not significantly different from the control mean (CB-F). Post feeding the treated group (STZ-P) has substantially greater mean than the control (CB-P). (B) Time course of glucose during 16 hours fasting. *X*-axis represents the time after removal of food when the glucose readings are taken and *Y*-axis represent the glucose levels. The grey band represents the upper and lower bounds of 95% CI of the control group with the mean glucose values represented by filled circles. Filled squares represent individuals that showed a monotonic decrease in glucose levels. In three animals the glucose levels reduced at or below the control levels and in two others they showed a continued monotonic decrease but did not reach the normal level in 16 h. Filled triangles with dotted lines represent the individual time courses of the three STZ treated rats which showed some indications of stabilising at a steady state above the normal.

liver glucose production (inlet tap) or the tissue glucose uptake (outlet tap), because of the presence of upper and lower set points, insulin resistance and the relative insulin insufficiency might change the time required to attain the steady state, but not the steady state itself (Fig. 2B). If CSS model is appropriate, insulin resistance or altered insulin levels are bound to change fasting glucose levels. The failure of insulin receptor knockouts and insulin suppression experiments to alter the fasting steady state, along with the delay

in reaching the steady state indicates that TSS model is likely to describe glucose homeostasis more appropriately. The TSS model requires mechanisms of sensing any departure from the targeted steady state. Such mechanisms are not known in peripheral systems but glucose sensing neurones are certainly known to be present in the brain. Therefore, if TSS is a more appropriate model, the CNS mechanisms are likely to be central to glucose homeostasis, particularly in determining the steady state levels; whereas insulin signalling would play a role in determining the rate at which a steady state is reached after perturbation.

### Testable predictions from the models

It is possible to make other testable predictions of TSS and CSS models. In the normal healthy individual, increased glucose utilisation is expected to decrease fasting glucose levels by the CSS model but not by the TSS model. Human experiments have shown that sustained exercise does not reduce plasma glucose, in fact it might increase (*Coggan, 1991*). In order to match with experimental data, CSS based models of glucose dynamics during exercise need to include additional terms which involve neuronal mechanisms such as direct stimulation of liver glucose production in response to exercise through sympathetic route (*Roy & Parker, 2006*). This brings the model close to a TSS model. If TSS model describes glucose homeostasis more appropriately, reduced insulin signalling is not expected to change steady state glucose but only alter the time course to reach a steady state.

The mechanism of attaining a hyperinsulinemic normoglycemic prediabetic state is different by the CSS and TSS models. By the classical CSS based pathway, obesity induced insulin resistance is believed to be primary. The insulin resistance reduces glucose uptake and the excess glucose triggers a compensatory insulin response. The resultant hyperinsulinemia compensates for insulin resistance keeping the fasting glucose levels normal. Detailed analysis of the model and matching its prediction with empirical data has refuted this model (*Chawla et al., 2018*). One of the intuitively appealing reasons for this refutation is that after the heightened insulin levels normalise glucose, there is no reason why insulin levels remain high. Therefore, a steady state with hyperinsulinemia and normoglycemia is impossible by the CSS model but it exists in a prediabetic state. If a "compensatory" insulin response is mediated by glucose, one would expect a positive correlation between fasting glucose (FG) and fasting insulin (FI) and no correlation between insulin resistance and β-cell responsiveness.

By the TSS model, on the other hand, compensatory response is possible in either way. Primary insulin resistance may increase the glucose levels transiently, but when glucose sensing mechanisms detect the change a compensatory response can be operational. By this mechanism a hyperinsulinemic normoglycemic state is possible. Alternatively, primary hyperinsulinemia (*Corkey, 2012*; *Garvey, Olefsky & Marshall, 1986*; *Shanik et al., 2008*; *Weyer et al., 2000*) can also be compensated by increased insulin resistance by hitting the lower level of sensing which would trigger compensatory insulin resistance. Even in this case a hyperinsulinemic normoglycemic state is possible. Both glucose sensing neurones and neuronal regulation of insulin release and liver glucose production are well

known. In the compensatory response mediated by TSS pathways there need not be a correlation between fasting insulin and fasting glucose, but insulin resistance and β cell response would be correlated.

Also using a simple CSS model (see Supplemental Information 2 for details), simulations show that the correlation coefficient and regression slope in the insulin-glucose relationship would remain the same in the fasting as well as post-meal state although the range of glucose and insulin levels will be different. On the other hand, in a TSS model the post-meal glucose and insulin levels are expected to be correlated but the steady state levels may not. We test these predictions by the two alternative models using human epidemiological data below.

We argued above that since on impairment of insulin signalling, the time required to reach a steady state can be substantially longer, overnight fasting may not ensure a steady state in all individuals. Fasting hyperglycaemia in T2D can have two alternative (but not mutually exclusive) causes. Either it represents the failure to reach a steady state in the specified fasting period, or it is because of mechanisms other than reduced insulin action. The TSS model can make differential predictions from the two alternative causes since it predicts a positive correlation between plasma glucose and plasma insulin in the post-meal state but loss of this correlation on reaching a steady state. In population data, if some individuals have reached a steady state but a few others haven't we would expect a correlation significantly weaker than the post-meal correlation. These predictions can be tested in epidemiological data.

## Analysis of insulin-glucose relationship in steady and perturbed state in human data: Epidemiological inquiry

In all three data sets there was weak ($R^2$ range 0.017–0.057) but significant correlation between fasting glucose (FG) and fasting insulin (FI) and strong correlation between HOMA-IR and HOMA-β ($R^2$ range 0.20–0.83) (Fig. 11).

It was seen in all three data sets that the correlation coefficients for glucose and insulin were an order of magnitude higher in the post-meal cross sectional data than in the fasting state (Table 7; Fig. 11). Also, the regression slopes in the post-meal data were substantially different from fasting data unlike what is expected by the CSS model (Supplemental Information 2). By both sets of predictions the CSS model predictions are rejected. The HOMA-IR HOMA-β correlation, as well as the difference between the regression correlation parameters between fasting and post-meal data are compatible with predictions of the TSS model. However, although weak, there is significant correlation between FG and FI unlike what may be expected by a steady state TSS model. This incompatibility is not sufficient to falsify the TSS model since the failure of a small proportion of individuals to have reached a steady state at overnight fasting is sufficient to explain the weak correlation. It is also likely that the assumption of fasting may not be true for the entire sample. Even if a small number of individuals do not comply with the overnight fasting instructions, a positive correlation can result and this possibility is extremely difficult to exclude in human data.

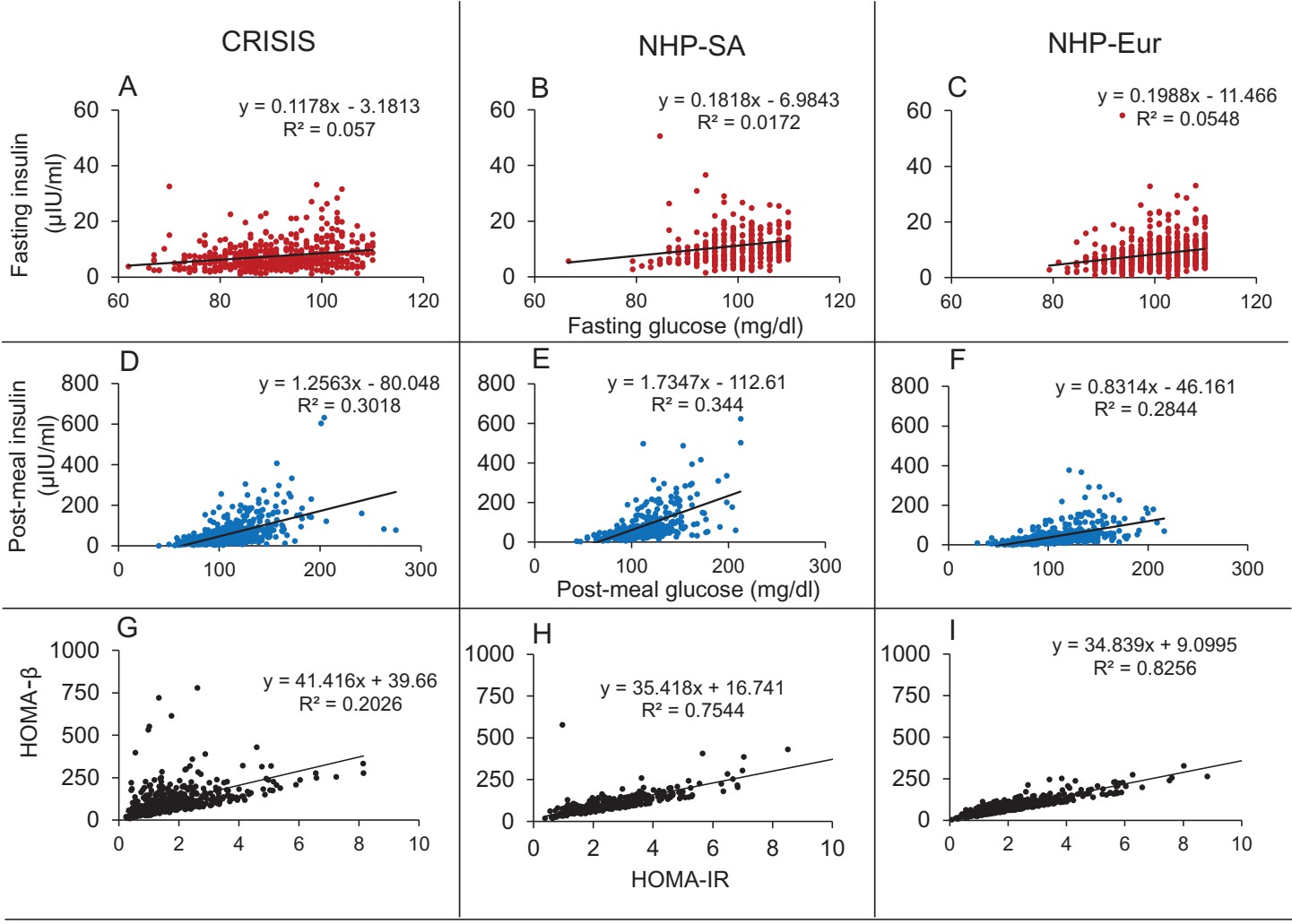

**Figure 11 Correlations in the human data.** The Fasting Glucose-Fasting Insulin (A–C), Post-meal Glucose-post-meal Insulin (D–F) and HOMA-IR HOMA-β (G–I) scatter plots in non-diabetic populations in the three data sets. The FG-FI correlation is weak as compared to post-meal correlation. The HOMA-IR and HOMA-β correlations are very strong in all the three data sets, which is not expected by the classical insulin resistance theory.

The support of TSS model over CSS model is important because it accounts for the failure of impairment of insulin signalling to alter fasting glucose but increase only post-load glucose.

## DISCUSSION

The five approaches examined above fail to support the classical belief about glucose-insulin relationship. The insulin receptor knock-out experiments and insulin suppression or enhancement experiments converge to show that alteration in insulin levels or insulin sensitivity does not change the steady state glucose levels. Evidence that it changes the shape of the glucose curve after food intake or glucose loading is more convincing in spite of some inconsistency across different experiments. Typically return to the steady state is

**Table 7 Correlation and regression parameters of glucose-insulin relationship at steady and perturbed states from human data.**

| Parameter → Data set ↓ | Steady state (fasting) | | | Perturbed state (2 h post glucose bolus) | | |
|---|---|---|---|---|---|---|
| | R-squared (variance explained) (%) | p Value | Slope (95% CI) | R-squared (variance explained) (%) | p Value | Slope (95% CI) |
| CRISIS (N = 522) | 0.0570 (5.7) | <0.0001 | 0.1178 [0.0765–0.1591] | 0.3018 (30.18) | <0.0001 | 1.2563 [1.0917–1.4209] |
| NHP-South Asian (N = 310) | 0.0172 (1.72) | 0.021 | 0.1818 [0.0279–0.3356] | 0.344 (34.4) | <0.0001 | 1.7347 [1.4661–2.0033] |
| NHP-European (N = 574) | 0.0548 (5.48) | <0.0001 | 0.1988 [0.131–0.2666] | 0.2844 (28.44) | <0.0001 | 0.8314 [0.7231–0.9397] |

delayed by impaired insulin signalling but the steady state glucose level remains unchanged. Convergence of experiments using other means of causing specific alterations in insulin action strengthens the inference.

A number of mathematical models attempt to capture the dynamics of glucose homeostasis. A good model should be able to explain all the empirical results summed up here namely the inability of insulin receptor knock-outs, insulin suppression and insulin enhancement experiments to alter steady state glucose levels; the difference in the regression correlation parameters between insulin and glucose in the steady versus perturbed state; the extremely weak correlation between fasting glucose and fasting insulin, but very strong correlation between HOMA-IR and HOMA-β; the hyperinsulinemic-normoglycemic prediabetic state and the phenomenon of impaired glucose tolerance but normal fasting glucose. Reviewing models of glucose homeostasis is beyond the scope of this paper, but we outline here what a good model of glucose homeostasis needs to explain. In our observation, all existing models explain only some of the empirical findings. We suggest here that this inability is because of a questionable common baseline assumption of all models that insulin signalling determines the glucose level in the fasting as well as post-feeding conditions. It should be possible to construct a model which is compatible with all experimental and epidemiological findings, if we realise that insulin affects glucose only in the post feeding but not in fasting conditions.

It is difficult to defend the classical assumptions about glucose-insulin relationship against the multiple convergent lines of evidence. Although results of these experiments have been there in the published literature for about two decades, these results were mostly explained away giving different excuses for different sets of experiments. The possible lines of defence would include difference between homeostatic mechanisms in rodents and humans or the possibility of non-linear nature of glucose-insulin relationship. The evidence reviewed here comes from rodents as well as humans and the glucose-insulin scatters do not show any clear indication of non-linearity. Further it would be prudent to avoid making inferences based on dietary or other complex interventions since they can have multiple mechanisms of action. Specific genetic or molecular interventions are more revealing with respect to the underlying mechanisms since we can be more confident about their specificity of action. Therefore our inference that insulin action does not influence fasting glucose levels is the most straightforward and parsimonious inference. Any other explanations will have to be supported by giving evidence for the assumptions made in those explanations.

The failure of experimental alteration in insulin signalling to alter steady state glucose raises two distinct possibilities about fasting hyperglycaemia in T2D. One is that fasting hyperglycaemia in T2D is a result of processes independent of insulin signalling such as autonomic signalling or other insulin independent mechanisms. The sympathetic tone is known to be altered in metabolic syndrome (*Thorp & Schlaich, 2015*) and increased sensitivity of liver to sympathetic signal is likely to be mainly responsible to fasting hyperglycaemia (*Bruce et al., 1992*). The other possibility is that with impaired insulin signalling overnight fasting is not sufficient to reach a steady state, therefore fasting hyperglycaemia in T2D is a non-steady state phenomenon in type 2 diabetes. The considerably weaker but still significant correlation between glucose and insulin in fasting as compared to post glucose load data suggests that both the factors are likely to be operational differentially in different individuals.

In either case certain fundamental concepts in our understanding of T2D need to be revised. First of all, the definition and measurement of insulin resistance using steady state glucose and insulin levels needs to be questioned. Most commonly used indices of insulin resistance are based on the assumption that insulin signalling decides the fasting steady state glucose levels, although non-equilibrium methods of assessing insulin resistance have been described (*Patarrão et al., 2012*). In the classical view other mechanisms of glucose regulation are assumed to be absent or non-significant. If increased sympathetic signalling increases liver glucose production, HOMA-IR will still account it as "insulin resistance". The same is true about insulin resistance measured by hyperinsulinemic euglycemic clamp. The way insulin resistance is measured at the clinical level eliminates the chance of separately accounting for other mechanisms of glucose regulation. Even when experiments show that certain agents affect glucose dynamics independent of insulin action, they are typically labelled as "insulin sensitising" agents (*Hossain et al., 2018*). As a result, the belief that insulin is the only mechanism of glucose regulation relevant to T2D is artificially strengthened. There is a subtle circularity in the working definition of insulin resistance. Insulin resistance is blamed for the failure of normal or elevated levels of insulin to regulate glucose. In order to test this hypothesis, we should have an independent definition and measure of insulin resistance. Only then we can test whether and to what extent insulin resistance can alter glucose dynamics. However, clinically insulin resistance is measured by the inability of insulin to regulate glucose. Such a measure cannot be used to test the hypothesis that insulin resistance leads to the failure of insulin to regulate glucose. The unfalsifiability of the insulin resistance hypothesis arising out of this circularity has halted any attempts towards realistic assessments of the true causes of fasting hyperglycaemia in type 2 diabetes. In the molecular approach to induce insulin resistance, we have an independent definition and causality for insulin resistance and therefore such experiments are free from circularity of definition. The results of such experiments reviewed here are therefore more revealing and reliable. Since all of them converge to show that altering insulin signalling does not alter steady state glucose levels, the insulin resistance and inadequate compensation hypothesis for steady state hyperglycaemia stands clearly rejected.

The question can be turned upside down to examine whether steady state glucose level determines steady state insulin. If glucose is infused with a constant rate over a long time, insulin levels will come back to the baseline levels if glucose is not a determinant of fasting insulin. If it is, then insulin levels will stabilise at a new heightened steady state level. *Jetton et al. (2008)* infused intra venous glucose (20% glucose w/v) continuously for 4 days in rats. Both glucose and insulin levels increased significantly after the infusion. However, later both glucose and insulin levels came back to normal even as the infusion continued. Increase in the concentration of the infused glucose (up to 35%) also yielded similar results (*Steil et al., 2001*). Thus, immediately on perturbation, glucose affected insulin levels, however after allowing sufficient time to regain steady state, the continued infusion of glucose had no significant effect on insulin levels. This demonstrates that even glucose does not hold a causal relationship with insulin in a steady state whereas glucose level perturbation is certainly known to stimulate insulin response.

The evidence reviewed above indicates that insulin is a driver but not a navigator of glucose homeostasis. A non-zero level of insulin is required for reaching a homeostatic steady state. In type 1 diabetes, the almost complete absence of insulin prevents glucose homeostasis. In type 2 diabetes there are non-zero insulin levels and therefore, a steady state is possible, but insulin itself plays little role in deciding the steady state glucose level. It is more likely that neuronal and other hormonal-metabolic factors affect the steady state glucose in T2D.

## CONCLUSIONS

From the insulin-glucose lesson we can learn certain general principles and norms for making causal inferences from experimental results. There are certainly more examples and implications of the driver navigator distinction than the ones captured by the insulin example. We see these phenomena in simple physico-chemical systems as well as in complex biological systems. An azeotropic mixture or a constant boiling solution has a steady state proportion of two or more liquids. If one starts from a non-equilibrium mixture, the rate of heating affects the time required to reach the azeotropic equilibrium but does not affect the equilibrium composition. This is because the molecular dynamics that decides the equilibrium has more complex elements than simple evaporation (*Li et al., 2020*) It is possible that in a given system some causal factors affect both steady and perturbed states where as others affect only one of them. For example, stirring increases the rate of dissolution of a solute, but does not affect the saturation concentration. Heating, on the other hand, can affect both.

Some of the well studies examples in complex systems include the regulation of plasmid numbers in bacteria. The rate of replication of plasmids does not decide the stable plasmid copy number since there is a complex set of mechanisms that decide the steady state independent of the rate of DNA replication (*Watve, Dahanukar & Watve, 2010*).

The differences between driver and navigator causes may also be specific to the context. For example, the rate of lexicon building at an early age may or may not be determining the adult vocabulary and language skills depending upon what is being tested (*Hill et al., 2017*; *Hurtado, Marchman & Fernald, 2008*; *Peter et al., 2019*)

In warm blooded animals, in a healthy state, the body temperature is maintained constant by homeostatic mechanisms. If the environmental temperature changes suddenly, heat transfer between the body and the environment is inevitable and as a result the body temperature will change transiently. However, the homeostatic loops will be operative and bring the body temperature back to normal soon. If an experimenter measures the body temperature immediately after changing the environmental temperature, a causal relationship might be apparent. If the body temperature is measured after reaching the steady state, one may infer that environmental temperature has no causal relationship with body temperature. If the experimenter is measuring heat transfer as the mechanism of causation, then it will be demonstrable at any time. So, a causal mechanism exists all the time but demonstration of a causal mechanism or pathway is not sufficient to establish a causal role.

These set of examples demonstrate that causation may work differently in steady versus perturbed states and the need to differentiate between drive and navigator causality is widespread in different fields of science. There is a need for a detailed philosophical account on this issue as well as a set of working norms for practising science. We suggest a few norms specifically for the field of experimental physiology.

Certain kinds of experimental interventions are unable to distinguish between driver versus navigator causality. Knocking out a driver or a navigator will disable the journey to the destination. Therefore, complete knockout of a cause may not distinguish between driver and navigator causality. On the other hand, experiments quantitatively altering the level of the causal factor while keeping it non-zero has a different set of implications. If a perturbation is momentary or transient, the results obtained would certainly reflect perturbed state causality, but may not reflect steady state causality. On the other hand, sustained perturbations held constant for sufficiently long to allow the system to regain a steady state are necessary to establish steady state causality. If upon sustainably altering a causal factor the effect variable returns to the same steady state, it reflects only perturbed state and not steady state causality. If, on the other hand, sustained alteration in the causal factor results into an altered steady state, it indicates steady state causality. A sub-normal driver will delay the time to destination but will not change the destination. On the other hand, changing the navigator may or may not alter the time, but will alter the position of the destination. In the history of insulin research, early experiments such as total pancreatectomy demonstrated the necessary role of insulin in glucose homeostasis but the distinction between driver or navigator causality was not even conceptually perceived. So, it was assumed that insulin does both the roles. Although the absence of correlation between fasting glucose and insulin but good correlation after perturbation was noted as early as 1969 (*Goodner, Conway & Werrbach, 1969*) in the absence of conceptual differentiation between steady state and perturbed state causality, a clear interpretation did not emerge. Now in the presence of multiple experiments showing the precise role of insulin, we need to revive our concepts of causality. Although the experiments reviewed above have been there in the literature, the inability of insulin to alter fasting glucose was not appreciated before, only because the distinction between

driver and navigator causality was not there in the philosophical foundation of biological causality.

For any homoeostatic system in which there can be one or more feedback loops working as homeostatic mechanisms and one or many possible perturbations, all the perturbations can have a demonstrable causal role in a non-equilibrium state but only some of the components of the feedback loops may have steady state causal relationship with the controlled parameter. If there are more than one homeostatic mechanisms operating on a controlled variable, it is possible that components of any one loop do not affect the steady state level of the variable. This distinction is important in building the philosophy of experimental physiology and medicine. Most of the experiments in physiology are perturbation experiments and often we try to understand the steady state in a homeostatic system using such experiments. This situation warrants care in making inferences in the absence of which our understanding of physiology can remain flawed.

## ACKNOWLEDGEMENTS

We would like to thank Prof. Raj Bhopal, University of Edinburgh for providing clinical data from the Newcastle Heart Project. We would also like to thank Prof. ChittaranjanYajnik, KEM Hospital, Pune for providing the clinical data from the CRISIS study. We are immensely grateful for the critical and insightful comments they provided on the earlier version of the manuscript. We would also like to thank Geetanjali Nerurkar for her help during the animal experiments.

### Funding

This work was supported by the Indian Institute of Science Education and Research, Pune, India. The funders had no role in study design, data collection and analysis, decision to publish, or preparation of the manuscript.

### Grant Disclosures

The following grant information was disclosed by the authors:
Indian Institute of Science Education and Research, Pune, India.

### Competing Interests

The authors declare that they have no competing interests.

### Author Contributions

- Manawa Diwekar-Joshi performed the experiments, analysed the data, prepared figures and/or tables, and approved the final draft.
- Milind Watve conceived and designed the experiments, analysed the data, authored or reviewed drafts of the paper, and approved the final draft.

## Animal Ethics

The following information was supplied relating to ethical approvals (i.e., approving body and any reference numbers):

Institutional Animal Ethics Committee (IEAC) of the Indian Institute of Science Education and Research (IISER) Pune approved the research based on the guidelines of Committee for the Purpose of Control and Supervision of Experiments on Animals (CPCSEA), India. Protocol Number IISER/IAEC/2016-02/006.

## Data Availability

The raw data are available in the Supplemental Files.

## Supplemental Information

Supplemental information for this article can be found online at http://dx.doi.org/10.7717/peerj.10396#supplemental-information.

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
