# Peer review of "Driver versus navigator causation in biology: the case of insulin and fasting glucose"

_PeerJ, doi:10.7717/peerj.10396_

## Round 0.1 · original submission · Minor Revisions

Please address critiques of both reviewers and amend your manuscript accordingly.

Reviewer 1 ·

Basic reporting

The study by Joshi and Watve highlights the possible misinterpretations of experimental data in biomedicine due to lack of the understanding of driver versus navigator causation. As a case study, they analyzed the glucose-insulin relationship to provide a potential proof of concept.
Authors have systematically analyzed the published data/literature and performed the relevant experiments to revisit the glucose-insulin relationship and highlighted the previously misinterpreted and exaggerated roles of insulin in regulating blood glucose levels in humans.

Experimental design

The study is novel, rigorously performed and systematically presented. The methodology and search criteria used are relevant and the limitations are also clearly mentioned which makes the study more reliable. Sufficient evidences from the literature and the experimental data supports the claim of the study. The manuscript is written in a lucid and comprehensive way.

Validity of the findings

However, some minor concerns must be addressed before publication.
1. References are missing at some places throughout the manuscript (for example in paragraph 246- 255, lines 471 and 473)
2. Figure No 8 and 9 are not sufficiently described and labeled. The figure legend should be written more comprehensively. It is difficult to understand what is shown in figure from the legend. More detail should be added in figure legends for figure no. 3-7.
3. Authors claim that fasting glucose in FIRKO knockouts remain unchanged, however some studies does not support this observation. Authors should carefully review these studies in detail to find the possible reason of inconsistency towards the conclusion of their original result.
4. To generalize the concept of driver and navigator causation in biology and/or in a broader prospective, some more cases with similar observation from the literature should be added (with references) in the discussion.,
5. Authors cite one hypothetical case of body temperature in the conclusion section, but this not supported relevant references.
6. Figure is not cited in result 2.5 (line no. 576)

·

Basic reporting

The author is trying to say there are logical traps in interventional inferences because steady state and perturbed state are different. They concluded that it is necessary to differentiate between types of causalities. To demonstrates this, they used the insulin and glucose as an example and reached the conclusion that researchers may have to carefully re-examine causal inferences from perturbation experiments and set up revised norms for experimental design for causal inference. The paper is well written and reads well, the background information is adequate. The result is clear and informative.
The figures are clear, and the subtitles are detailed. The reasonings in the paper are also clear.
However, there are some minor parts that needs to be improved.
1. Figure 2 is a great analogy, the author explained the analogy between 2A and insulin sensitivity, it would be great if they also explain the analogy between 2B and the glucose homeostasis model

Experimental design

The experiment design is good. The question is well defined and then the author used 5 different approaches to prove the concepts.

Validity of the findings

The conclusion are well-stated. The reasoning is also reasonable and clear.

---

## Round 0.2 · accepted · Accept

Thank you for your response to the reviewers' critiques and revised manuscript. I am pleased to inform you that your manuscript has been accepted for publication.